# RoboFlow4D: A Lightweight Flow World Model Toward Real-Time Flow-Guided Robotic Manipulation

Sixu Lin [1] [*]   Junliang Chen [2] [*]   Huaiyuan Xu [2] [†]   Zhuohao Li [3]   Guangming Wang [4]   Yixiong Jing [4]   Sheng Xu [1]
Runyi Zhao [1]   Brian Sheil [4]   Lap-Pui Chau [2]   Guiliang Liu [1] [3] [†]

## Abstract

Planning and acting in 3D environments is a fundamental capability for robotic manipulation in the real world. Although prior work has explored predictive flow planners to guide 3D manipulation, existing approaches often rely on modular pipelines stacking multiple submodels, resulting in high computational overhead and limited real-time performance. To address these challenges, we introduce RoboFlow4D, a lightweight flow world model that unifies perception and planning by estimating temporal motion in physical 3D space. As an end-to-end framework, RoboFlow4D directly predicts multi-frame 3D flows from visual observations and textual instructions, providing explicit flow-based planning to guide action generation. This design allows seamless integration with general action policies, forming an efficient observation–planning–execution closed loop. Through slow–fast collaboration between flow prediction and action control, RoboFlow4D enables real-time and resource-efficient manipulation. Extensive experiments in both simulation and real-world settings demonstrate that RoboFlow4D consistently improves manipulation success rates and computational efficiency, advancing flow-guided planning for embodied intelligence. Our project page is available at `RoboFlow4D`.

## 1. Introduction

As an important cornerstone toward developing embodied generalist agents, recent learning-based manipulation ap-

---
[*]Equal contribution   [†]Corresponding authors.   [1]School of Data Science, The Chinese University of Hong Kong (Shenzhen) [2]The Hong Kong Polytechnic University [3]Shenzhen Loop Area Institute [4]University of Cambridge. Correspondence to: Huaiyuan Xu <huaiyuan.xu@polyu.edu.hk>, Guiliang Liu <liuguiliang@cuhk.edu.cn>.

*Proceedings of the 43rd International Conference on Machine Learning*, Seoul, South Korea. PMLR 306, 2026. Copyright 2026 by the author(s).

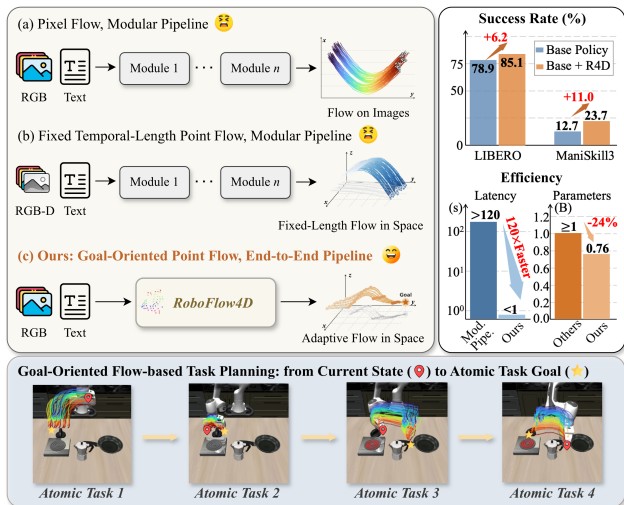

*Figure 1.* **Top left**: System-level comparison of various flow-based planning. (a) 2D flow-based planning (Vecerik et al., 2024; Xu et al., 2024) predicts pixel-level flow on images using a modular pipeline with stacked modules, but lacks 3D geometry. (b) Point flow-based frameworks (Li et al., 2025a; Dharmarajan et al., 2025) improve pixel flows by typically predicting fixed temporal-length 3D flows from 3D observations (*i.e.*, RGB-D images), while still relying on a modular pipeline. (c) The end-to-end flow-based pipeline designs a unified world model that directly predicts multi-frame flows across time and 3D space (*i.e.*, 4D spacetime) from RGB and text inputs. **Bottom**: Our RoboFlow4D adaptively adjusts the flow planning horizon from the current state to the atomic task goal. **Top right**: RoboFlow4D improves the success rate over the base policy by >6.2% in both LIBERO (Liu et al., 2023) and ManiSkill3 (Tao et al., 2024) simulations. Furthermore, RoboFlow4D achieves 120× speedup over modular pipelines and reduces model scale by >24% compared to other flow models.

proaches aim to build end-to-end models that generate executable robot actions directly from visual observations and textual instructions (Brohan et al., 2022; Hou et al., 2024; Black et al., 2026). This *observation → action* paradigm enables a wide range of general-purpose skills such as grasping, pushing, and stacking (Liu et al., 2024a; Kim et al., 2024; Chi et al., 2023). Although this paradigm has demonstrated substantial effectiveness, it still encounters unknown failures. To address this limitation, incorporating pre-execution planning as an explicit guiding signal is crucial for achieving robust robotic manipulation (Xu

et al., 2024; Gao et al., 2024; Ji et al., 2025; Yu et al., 2025; Dharmarajan et al., 2025).

An intuitive way to instantiate the *observation → planning → action* paradigm is to i) plan future manipulation trajectories (also referred to as flows (Dharmarajan et al., 2025; Huang et al., 2026)) from observations and instructions, and ii) guide action generation by tracking the planned trajectories. Following this paradigm, recent works typically exploit world models (LeCun, 2022) to plan in image space (Gao et al., 2024; Bharadhwaj et al., 2024; Yu et al., 2025; Fan et al., 2025; Yang et al., 2025a). They predict 2D flows to represent scene evolution (Bharadhwaj et al., 2024), object movement (Xu et al., 2024), or robotic arm motion (Wen et al., 2023). However, guiding robot manipulation in the 3D world using these 2D flows is inherently ill-posed. Pixel-level trajectories defined in image space lack crucial spatial awareness, such as depth and geometry in the 3D environment. As a result, a "seemingly reasonable" 2D trajectory may lead to collisions, near-misses, or physically infeasible motions (Li et al., 2025b; Dharmarajan et al., 2025).

Inspired by humans' ability in 3D dynamics modeling (Ai et al., 2025), it is also essential for the embodied robotic planners to generate 3D trajectories. To this end, (i) One possible approach is to merge 2D image trajectories with depth estimations to form pseudo-3D flows (Zhi et al., 2025). (ii) Another line of work aims to generate genuine 3D trajectories (Ye et al., 2025; Dharmarajan et al., 2025). This approach first predicts future scene evolution using a video generation expert (Wan et al., 2025; Wiedemer et al., 2025), and then integrates specialized expert modules (e.g., depth estimation (Yang et al., 2024; Chen et al., 2025), grounding (Liu et al., 2024b), and point tracking (Karaev et al., 2025)) to reconstruct realistic 3D flows. However, these solutions typically rely on *stacking multiple third-party components* (Ni et al., 2025; Ardelean et al., 2025). Although such a highly modular and component-dependent pipeline reduces research and development (R&D) complexity, *stacking multiple expert modules inevitably introduces substantial computational and memory overhead* (Hu et al., 2023). This burden makes it impractical to deploy on robots operating in real time, where low-latency, high-efficiency, and lightweight models are required.

To enable real-time robotic deployment, we propose **RoboFlow4D**, an end-to-end lightweight world model that directly predicts a sequence of multi-frame 3D flows (i.e., flows across 4D spacetime), conditioned on RGB images and textual instructions. RoboFlow4D possesses two main advantages: (i) It significantly improves the inference efficiency over the modular pipeline; (ii) It can serve as a *plug-and-play lightweight* guiding module for downstream policies to produce manipulation actions.

By leveraging **RoboFlow4D**, we develop an efficient closed-loop robotic manipulation framework. Unlike the traditional cascaded planning–control architecture (Xu et al., 2024; AgiBot-World-Contributors et al., 2025), our framework adopts a dual-system architecture enabling slow-fast collaboration (Kahneman, 2011; Shi et al., 2025; Bjorck et al., 2025). Its high efficiency is ensured in three aspects. (1) **Lightweight networks**: Both the flow world model and the policy are lightweight, therefore improving overall framework efficiency; (2) **A goal-oriented flow world model**: RoboFlow4D adaptively adjusts the time span required to complete an atomic task from the current state, which improves planning efficiency compared to predicting a fixed time span; (3) **Slow-fast collaboration**: RoboFlow4D acts as the planner, while the action policy serves as the executor, with planning and execution running at different frequencies to reduce latency. Extensive experiments in both simulation and real-world settings demonstrate that RoboFlow4D improves manipulation success rates over the base policy, while maintaining low planning latency and a lightweight model scale (top right in Figure 1). The advancement largely accelerates task completion (Table 5). The contributions of the paper are summarized as follows:

- An end-to-end efficient flow world model: RoboFlow4D directly predicts multi-frame 3D flows from 2D images and text instructions using a unified pipeline, thereby eliminating the high computational cost of modular pipeline stacking expert modules.
- A novel, efficient plug-and-play 3D flow-guided policy learning paradigm: Lightweight RoboFlow4D produces 3D plans to guide action learning with a low-cost policy.
- Slow-fast closed-loop manipulation: A closed loop of observation–planning–execution coordinates low-frequency goal-oriented planning with high-frequency control, yielding efficient and accurate manipulation.
- Extensive empirical validation in simulation and real-world settings: RoboFlow4D consistently improves success rates across LIBERO/ManiSkill3 simulations by 6.2%/11.0% and real-world tasks by 5-20%, and offers <1 second low-latency planning. Moreover, flow-guided manipulation yields reduced task completion time.

## 2. Related Work

### 2.1. 2D Trajectory-Guided Manipulation

2D trajectory–guided manipulation explores an image-space planning interface for robotic control, using scene-centric, object-centric, or gripper-centric 2D pixel-level trajectories. Scene-centric approaches, such as Track2Act (Bharadhwaj et al., 2024), predict 2D trajectories of the entire scene from initial and goal images, after which a control policy is learned conditioned on these predicted trajectories. Although scene-centric approaches can capture scene-level motions, distinguishing fine-grained object or gripper movements remains challenging. To this end, object-centric ap-

proaches including RoboTAP (Vecerik et al., 2024) and Im2Flow2Act (Xu et al., 2024) offer a practical manner to focus exclusively on the object of interest. Specifically, Im2Flow2Act samples query points from the grounded object in the initial frame via Grounding DINO (Liu et al., 2024b), and subsequently tracks these keypoints across subsequent frames using an off-the-shelf tracking expert (Doersch et al., 2023). Beyond modeling object-centric dynamics, gripper-centric methods, such as ATM (Wen et al., 2023), estimate gripper trajectories as conditions for policy learning. Although manipulation performance benefits from understanding object or gripper dynamics, these approaches rely on 2D image-plane trajectories, which inherently lack 3D geometric information. As an extension of 2D gripper-centric methods, our approach predicts 3D trajectories, enabling a better representation of robot motion in a 3D world.

## 2.2. 3D Spatially-Aware Action Modeling

In light of the inherent limitations of 2D trajectory-guided manipulation, recent 3D-aware approaches seek to incorporate 3D geometry to improve manipulation performance. These methods span both lightweight policies and heavy vision–language–action (VLA) models. Lightweight policies, such as DP3 (Ze et al., 2024), PointFlowMatch (Chisari et al., 2024), FP3 (Yang et al., 2025b), encode 3D point cloud observations to be 3D features, and then learn an action policy conditioned on these features. VLA models, such as SpatialVLA (Qu et al., 2025) and 4D-VLA (Zhang et al., 2025), inject 3D information via a 3D-aware position encoding strategy to model 3D space.

Additionally, 3D trajectory-based spatially-aware methods aim to predict motion in 3D space to guide action modeling. For example, 3DFlowAction (Zhi et al., 2025) augments 2D image trajectories (Karaev et al., 2025) with depth estimations (Yang et al., 2024) to derive pseudo 3D motion. In contrast, other approaches explicitly generate real 3D object trajectories for action modeling (Li et al., 2025a; Dharmarajan et al., 2025). These methods adopt a modular pipeline: (i) synthesizing future videos via video generation experts, (ii) estimating depth using video depth experts, and (iii) predicting 3D motion through 3D point tracking and grounding modules. Despite being effective, these approaches require substantial resource bandwidth of the onboard chip to stack expert modules, rendering them impractical for real-world robot deployment. Recently, PointWorld (Huang et al., 2026) has improved prediction efficiency with an end-to-end pipeline, which predicts fixed temporal-window flows in 3D space from RGB-D inputs and historical flow observations. In contrast, RoboFlow4D takes RGB images and textual instructions as input, and predicts goal-oriented multi-frame flows in 3D space with an adaptive temporal horizon, enabling task-adaptive prediction lengths.

## 3. Methodology

### 3.1. Overview

The overall robotic manipulation framework is illustrated in Fig. 2, comprising three main components: RoboFlow4D, Flow-Conditioned policy Learning, and Closed-Loop Control. Below is an overview of each component.

**RoboFlow4D.** The RoboFlow4D module is designed to predict future multi-frame flows in 3D space based on visual observations and textual instructions. RoboFlow4D adopts an end-to-end pipeline built upon a unified network, rather than stacking expert modules. Specifically, RoboFlow4D extracts visual, 2D point (optional), and textual features with their respective encoders. To inject 3D prior into the visual features, we design a 3D perceiver and distill 3D knowledge from a 3D foundation model (*e.g.*, VGGT (Wang et al., 2025)), resulting in 3D-aware features. Next, a proposed diffusion-based FlowDiT predicts flows by performing denoising conditioned on the extracted multimodal features.

**Flow-Conditioned policy Learning.** The multi-frame flows provided by RoboFlow4D offer strong motion guidance for robotic manipulation. Accordingly, a flow-conditioned action policy generates action chunks that are modulated by the current state (*i.e.*, the image observation and robot proprioception) and an explicit flow plan.

**Closed-Loop Control.** By coupling RoboFlow4D with an action policy as a slow-fast dual system, we instantiate a closed-loop control for robotic manipulation. Specifically, goal-oriented RoboFlow4D acts as a low-frequency flow planner, whose single-step plan extends beyond the time horizon of an action chunk. The action policy serves as a high-frequency executor, which rolls out multiple action chunks conditioned on the one-step flow plan. This closed-loop control enables the planner and executor to interact until the manipulation task is completed.

The following sections introduce the details of RoboFlow4D (Sec. 3.2), Flow-Conditioned Policy Learning (Sec.3.3), and Closed-Loop Control (Sec. 3.4).

### 3.2. RoboFlow4D

RoboFlow4D takes as input a sequence of historical RGB images $\mathcal{I} = \{I_1, I_2, \ldots, I_n\}$, optional 2D query points $\mathcal{Q} \in \mathbb{R}^{m \times 2}$, and a textual task instruction, where $n$, $m$ refer to the number of images and query points, respectively. We leverage a Vision Encoder, a Point Encoder, and a Text Encoder to extract the corresponding tokens of each input modality. Specifically, the Vision Encoder extracts local patch tokens $T_{\text{local}} \in \mathbb{R}^{nl \times C}$ and global image tokens $T_{\text{global}} \in \mathbb{R}^{n \times C}$ from the image sequence using DINO v2 (Oquab et al., 2023) and SigLip (Zhai et al., 2023), respectively. $l$ denotes the number of patch tokens per image, $C$

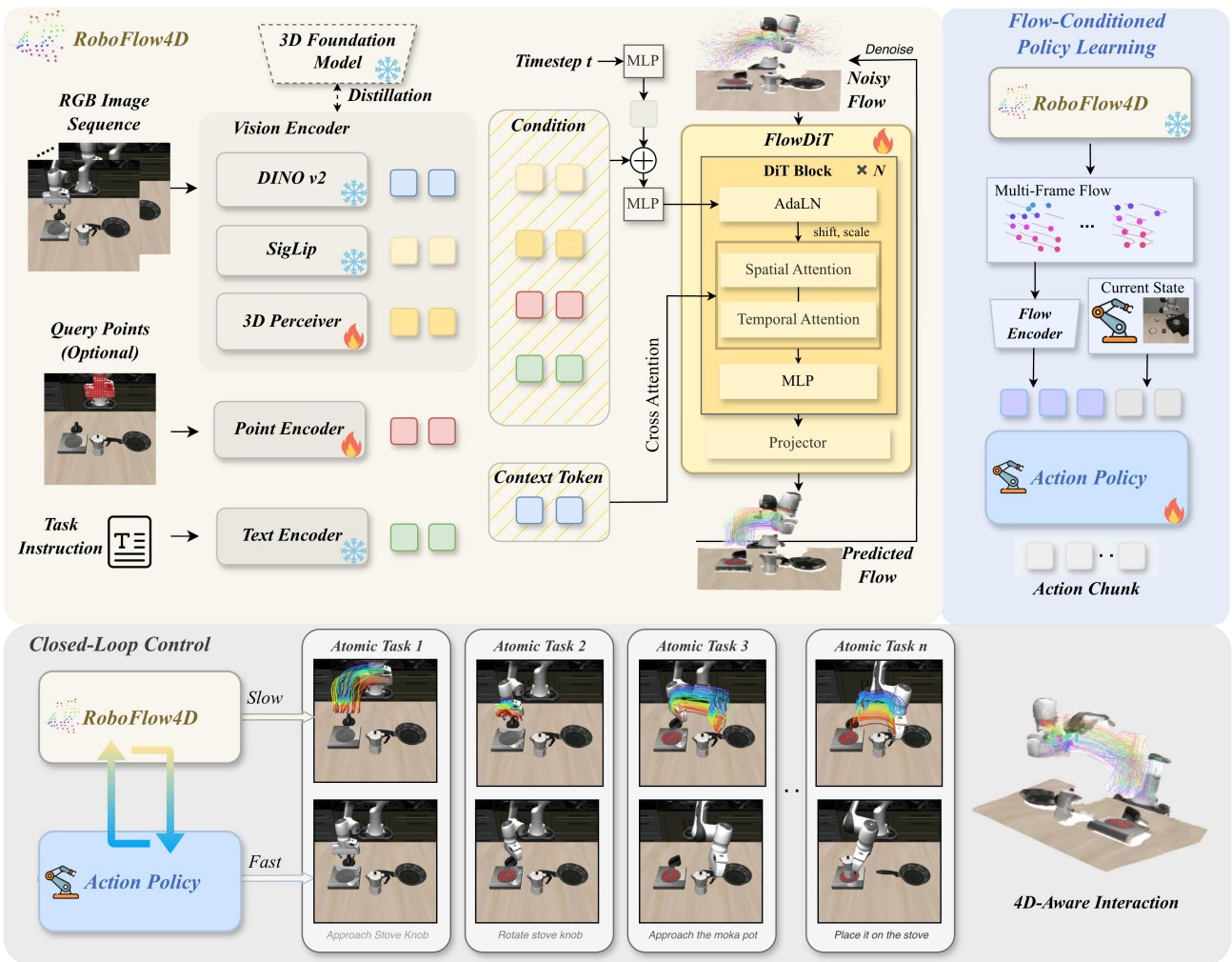

*Figure 2.* Overview of robotic manipulation. **Top left** (Sec. 3.2): Given an RGB image sequence, optional gripper query points, and a task instruction, the proposed RoboFlow4D extracts vision, 2D point, and text tokens using a Vision Encoder, Point Encoder, and Text Encoder, respectively. A diffusion-based FlowDiT predicts future multi-frame flows from the extracted tokens. **Top right** (Sec. 3.3): Built upon RoboFlow4D, an action policy learns to generate actions conditioned on the current state and explicit flow. **Bottom** (Sec. 3.4): By integrating RoboFlow4D with the action policy, we further develop an observation-planning-execution closed loop for efficient robotic manipulation, in which RoboFlow4D acts as a slow planner and the action policy serves as a fast executor.

is the dimension of each token. We then merge the global image tokens into $G \in \mathbb{R}^{1 \times C}$ via mean pooling and aggregate local patch tokens into context tokens $L \in \mathbb{R}^{n_{\text{local}} \times C}$ by applying a multi-head self-attention (MHA) (Vaswani et al., 2017) to a set of zero-initialized learnable queries $Q_{\text{local}}$:

$$L = \text{MHA}(\text{query=}Q_{\text{local}}, \ \text{key=}T_{\text{local}}, \ \text{value=}T_{\text{local}}), \quad (1)$$

where $n_{\text{local}}$ is the number of aggregated local tokens. The task instruction is encoded by the Text Encoder (*i.e.*, the text encoder of SigLip) into the text token $T_{\text{text}} \in \mathbb{R}^{1 \times C}$. For the optional 2D point input, the Point Encoder first projects them into point tokens $T_{\text{point}} \in \mathbb{R}^{m \times C}$ using a multi-layer perceptron (MLP), and then extracts positional information by aggregating the point tokens into a compact latent embedding using an MHA with a learnable query

token $q_{\text{point}} \in \mathbb{R}^{1 \times C}$:

$$Q = \text{MHA}(\text{query=}q_{\text{point}}, \ \text{key=}T_{\text{point}}, \ \text{value=}T_{\text{point}}), \quad (2)$$

where $Q \in \mathbb{R}^{1 \times C}$ denotes the aggregated latent embedding, which is set to zero when query points are not provided.

**3D Perceiver.** Understanding 3D geometry of the environment is crucial for robotic manipulation. To empower RoboFlow4D with 3D awareness from 2D observations, we design a 3D Perceiver that distills 3D knowledge from an off-the-shelf 3D foundation model, VGGT (Wang et al., 2025). The 3D Perceiver comprises a set of learnable 3D queries $Q_{\text{3D}} \in \mathbb{R}^{n_{\text{3D}} \times C}$ and a Resampler (Alayrac et al., 2022) with stacked cross attentions and feed-forward networks, where $n_{\text{3D}}$ refers to the number of 3D queries. We first use the Resampler to encode 3D geometry from context

tokens $L$ using the 3D queries, and then leverage an MLP to project the Resampler output into 3D-aware condition $T_{3D} \in \mathbb{R}^{1 \times C}$. By introducing an alignment loss between $T_{3D}$ and the mean-pooled features from VGGT, we inject 3D knowledge into the 3D condition $T_{3D}$.

**FlowDiT.** Built upon the extracted multimodal features, we develop a diffusion-based FlowDiT to effectively predict future flows conditioned on these features. The FlowDiT module consists of $N$ stacked diffusion transformer (DiT) (Peebles & Xie, 2023) blocks and an MLP Projector, where each block comprises adaptive layer norm (AdaLN), spatiotemporal MHA, and MLP operations. Specifically, we first construct a multimodal condition by concatenating $G$, $T_{3D}$, $Q$, and $T_{text}$ into a single token $T_{cond} \in \mathbb{R}^{1 \times 4C}$ along the channel dimension. The multimodal condition is augmented with timestep information encoded by MLP. Subsequently, noisy flows and the multimodal condition are jointly fed into DiT blocks for conditioned denoising. To enhance perceptual capability, we introduce a spatiotemporal cross-attention to replace the original MHA in each DiT block. Finally, the denoised flows obtained after iterative denoising steps are passed through the Projector to generate the final flow $\mathcal{F} \in \mathbb{R}^{n_{kp} \times K \times 3}$, where $n_{kp}$, $K$ are the number of keypoints and frames, respectively.

### 3.3. Flow-Conditioned Policy Learning

The flow plan $\mathcal{F} \in \mathbb{R}^{n_{kp} \times K \times 3}$ from RoboFlow4D is encoded by a Flow Encoder as a condition indicating robot motion for the action policy. Specifically, we first use an MLP to extract keypoint features $F_{kp} \in \mathbb{R}^{n_{kp} \times K \times C_{cond}}$:

$$F_{kp} = \text{MLP}(\mathcal{F}), \tag{3}$$

where $C_{cond}$ is the number of channels. The keypoint features $F_{kp}$ are aggregated across keypoints and frames using attention pooling $\text{AttnPool}(\cdot)$ to acquire a global flow condition $f_{flow} \in \mathbb{R}^{1 \times C_{cond}}$:

$$f_{flow} = \text{AttnPool}(F_{kp}). \tag{4}$$

The final condition $f_{cond} \in \mathbb{R}^{1 \times (n_{base}C + C_{cond})}$ is obtained by combining the global flow condition and the base condition $f_{base} \in \mathbb{R}^{1 \times n_{base}C}$ encoded from the current states (*i.e.*, vision-language and proprioception observations):

$$f_{cond} = [f_{base}; f_{flow}], \tag{5}$$

where $n_{base}$ is the number of tokens of the base condition.

RoboFlow4D is frozen to ensure stable flow plans during policy learning. We provide empirical studies to validate the general benefits of RoboFlow4D-guided action learning across various policies, including the Diffusion Policy (DP) (Chi et al., 2023) and the Diffusion Transformer Policy (DiT) (Hou et al., 2024).

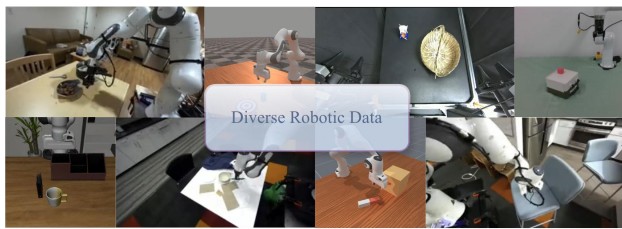

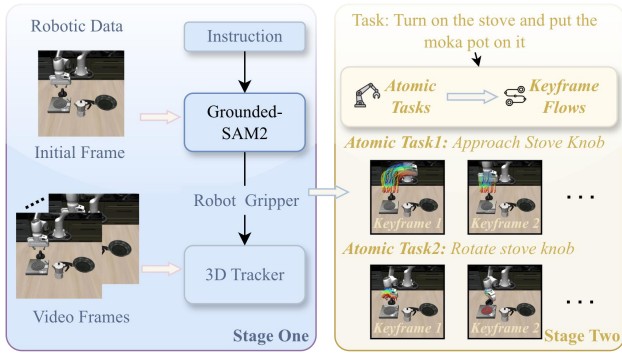

*Figure 3.* Data generation pipeline. **Stage one**: We track the flows of the grounded gripper from diverse real-world and simulated robot videos. (Khazatsky et al., 2024; Liu et al., 2023; Tao et al., 2024). **Stage two**: Each video is divided into sequential atomic tasks, and then goal-oriented flows are collected from a specific keyframe to each atomic task completion.

### 3.4. Closed-Loop Control

Open-loop control is ill-suited to manipulation due to compounded errors and unresponsiveness to robot and environmental deviations resulting from a single-step, one-shot policy. Hence, we establish a closed-loop control framework that enables the robot to observe, plan, and act iteratively across the manipulation task horizon. Each loop is accomplished by (1) RoboFlow4D producing an atomic task plan from the current observation and (2) a lightweight action policy executing all action chunks conditioned on the plan. New observations from the next loop are fed to RoboFlow4D after the atomic task completes.

The two systems are coordinated in an efficient dual-frequency scheme: goal-oriented RoboFlow4D plans the entire robot trajectory in a single step at a lower frequency, and the policy executes multiple action chunks guided by the plan at a lower frequency. Such dual-frequency slow–fast cooperation (one-step planning followed by multi-step execution) performs more efficiently than the same-frequency scheme (one-step planning followed by one-step execution).

### 3.5. Data Generation and Training Objective

**Data Generation.** To enable RoboFlow4D to plan 3D paths from 2D images, 3D flow supervision is introduced to equip it with 3D geometric awareness. The data generation pipeline (Fig. 3) comprises two stages from both real-world (Khazatsky et al., 2024) and simulated (Liu et al., 2023; Tao et al., 2024) environments: (1) grounding gripper to track

3D flows, (2) segmenting serial atomic tasks from videos and sampling keyframe flows of each atomic task.

Specifically, (i) we focus on the gripper's motion as it offers a strong prior for the robot arm in 3D space. We first sample $n_{\text{kp}}$ query points in the mask grounded by Grounded-SAM2 (IDEA-Research) on the robot end-effector (*i.e.*, the gripper) and then track the corresponding 3D points $\tilde{\mathcal{F}}_t \in \mathbb{R}^{n_{\text{kp}} \times 3}$ at time $t$ with SpatialTrackerV2 (Xiao et al., 2025).

(ii) A manipulation typically comprises sequential atomic tasks (*e.g.*, pre-grasp → grasp → transport → release). By binarizing the gripper open/close signal $g_t$ to obtain $b_t = \mathbb{I}[g_t > 0] \in \{0, 1\}$, the manipulation is divided into atomic tasks when a stable switch in $b_t$ is detected. We sample $K$ keyframes for each atomic task. More keyframes near the atomic task end are sampled to capture a more precise trajectory prior to when the gripper approaches a change. Keyframe selection follows a monotonic warping rule $u_k$:

$$u_k = \left( \frac{k}{K-1} \right)^{\gamma}, \quad t_k = \lfloor s_i + u_k (e_i - s_i) \rfloor, \quad (6)$$

where $s_i$ and $e_i$ denote the start and end time step of the atomic task. A value of $\gamma > 1$ allocates more keyframes near $e_i$. Additional details are provided in Appendix C.

**Training Objective.** RoboFlow4D is trained as a conditional denoising diffusion model (Ho et al., 2020) over 3D trajectories, with $v$-prediction parameterization (Salimans & Ho, 2022) to improve training stability. The overall objective comprises three losses: (1) a diffusion denoising loss $\mathcal{L}_{\text{diff}}$ that drives the model to recover physically plausible trajectories from noisy inputs; (2) an alignment loss $\mathcal{L}_{\text{align}}$ strengthening 2D-to-3D perception by aligning 3D representations between RoboFlow4D and VGGT (Wang et al., 2025); and (3) a smooth policy loss $\mathcal{L}_{\text{smooth}}$ ensuring temporal consistency by penalizing second-order differences of the denoised trajectories (Barron, 2019): $\mathcal{L} = \mathcal{L}_{\text{diff}} + \lambda_{\text{align}} \mathcal{L}_{\text{align}} + \lambda_{\text{smooth}} \mathcal{L}_{\text{smooth}}$, where $\lambda_{\text{align}}$ and $\lambda_{\text{smooth}}$ are scalar weights. More details are in Appendix D.

# 4. Experiments

## 4.1. Experimental Setup

**Simulation Benchmarks.** We evaluate our method on two widely recognized robotic benchmarks: (1) **LIBERO** (Liu et al., 2023), a lifelong learning benchmark with 5 suites spanning 130 tasks; *LIBERO-Spatial* evaluates spatial generalization by varying object placements; *LIBERO-Object* tests object generalization by placing different objects into a box; *LIBERO-Goal* measures diverse operations in a fixed environment. (2) **ManiSkill3** (Tao et al., 2024), a reliable robotic manipulation platform with diverse simulated environments and standardized metrics for evaluation. We specifically focus on three manipulation tasks: *PushCube*,

*PickCube*, and *StackCube*.

**Baseline.** We compare our approach against representative baselines across all evaluation settings. 1) LIBERO: various recent VLA models, including Octo (Team et al., 2024), CogACT (Li et al., 2024), OpenVLA (Kim et al., 2024), TraceVLA (Zheng et al., 2024), SpatialVLA (Qu et al., 2025) and 4D-VLA (Zhang et al., 2025). 2) ManiSkill3, including classical imitation and VLA baselines, such as BC (Mandlekar et al., 2021), ACT (Zhao et al., 2023) and OpenVLA (Kim et al., 2024). BC and ACT follow the official ManiSkill3 implementations, while OpenVLA follows the implementation of (Liu et al., 2024a).

**Implementation Details.** We implement DP (Chi et al., 2023) and DiT (Hou et al., 2024) and condition both policies on visual observations and proprioceptive inputs. For LIBERO, we follow the standard benchmark setup and use multi-view observations from a third-view camera and a wrist-mounted camera, training on the official demonstration datasets for fair comparison. For ManiSkill3, we use only the third-view image and collect 100 motion-planning demonstrations per task for training. Proprioceptive signals are included in all settings.

## 4.2. Main Results

**LIBERO.** Table 1 shows that RoboFlow4D-guided lightweight policies consistently benefit from explicit 4D motion prior. RoboFlow4D strengthens efficient controllers competitive with large VLA baselines. DP achieves substantial improvements in success rates of 8.2%, 8.0%, and 6.2% on *Spatial*, *Long*, and average, respectively. Analogously, on the same suites, DiT is significantly promoted to 90.2%, 75.2%, and 87.7%, respectively, which is better or competitive with many VLAs. On the easy *Object* suite, the lightweight policies still obtain considerable gains.

**ManiSkill3.** Table 2 demonstrates the consistent improvements for lightweight controllers from RoboFlow4D under the difficult single third-view setting. All baselines exhibit low success rates in such a difficult setting. By introducing the prior from RoboFlow4D, DP acquires 10.0%, 10.0%, 9.0%, 9.7% gains on *PushCube*, *PickCube*, *StackCube*, and average, respectively. RoboFlow4D further boosts DiT by $\geq 10.0\%$ across all tasks, resulting in an average improvement of 11.0%. These consistent improvements justify the superiority of 3D flow plan guidance for manipulation in challenging single-view scenarios with limited spatial cues.

The results above, across both LIBERO and ManiSkill3 simulations, validate the benefits of 3D flow plans from RoboFlow4D as an interface for downstream action generation using lightweight action policies.

*Table 1.* **Success rates (%).** Quantitative results of VLAs for fine-tuned robotic manipulation tasks on the LIBERO benchmark. **Best** results are in **bold** and **second-best** results are underlined.

| Method | Spatial | Object | Goal | Long | Average |
|---|---|---|---|---|---|
| Octo (Team et al., 2024) | 78.9 | 85.7 | 84.6 | 51.1 | 75.1 |
| CogACT (Li et al., 2024) | 87.5 | 90.2 | 78.4 | 53.2 | 77.3 |
| OpenVLA (Kim et al., 2024) | 84.7 | 88.4 | 79.2 | 53.7 | 76.5 |
| TraceVLA (Zheng et al., 2024) | 84.6 | 85.2 | 75.1 | 54.1 | 74.8 |
| SpatialVLA (Qu et al., 2025) | 88.2 | 89.9 | 78.6 | 55.5 | 78.1 |
| 4D-VLA (Zhang et al., 2025) | 88.9 | 95.2 | **90.9** | **79.1** | **88.6** |
| DP (Chi et al., 2023) | 81.6 | 91.5 | 78.4 | 64.0 | 78.9 |
| **w/ RoboFlow4D** | 89.8 | 93.2 | 85.2 | 72.0 | 85.1 |
| Δ | **+8.2** | **+1.7** | **+6.8** | **+8.0** | **+6.2** |
| DiT (Hou et al., 2024) | 84.2 | 96.3 | 85.4 | 68.8 | 83.7 |
| **w/ RoboFlow4D** | **90.2** | **97.0** | 88.4 | 75.2 | 87.7 |
| Δ | **+6.0** | **+0.7** | **+3.0** | **+6.4** | **+4.0** |

*Table 2.* **Success rates (%).** Each task is evaluated over 100 trials. **Best** results are in **bold** and **second-best** results are underlined.

| Method | Push | Pick | Stack | Avg. |
|---|---|---|---|---|
| BC (Mandlekar et al., 2021) | 26.0 | 1.0 | 0.0 | 9.0 |
| ACT (Zhao et al., 2023) | 30.0 | 2.0 | 1.0 | 11.0 |
| OpenVLA (Kim et al., 2024) | 10.0 | 0.0 | 2.0 | 4.0 |
| DP (Chi et al., 2023) | 31.0 | 3.0 | 3.0 | 12.3 |
| **w/ RoboFlow4D** | 41.0 | 13.0 | **12.0** | 22.0 |
| Δ | **+ 10.0** | **+ 10.0** | **+ 9.0** | **+ 9.7** |
| DiT (Hou et al., 2024) | 32.0 | 4.0 | 2.0 | 12.7 |
| **w/ RoboFlow4D** | **45.0** | **14.0** | **12.0** | **23.7** |
| Δ | **+ 13.0** | **+ 10.0** | **+ 10.0** | **+ 11.0** |

## 4.3. Ablation Study

**Modular Ablation.** Table 3 validates the effectiveness of each design in RoboFlow4D, measured by 3D $\ell_2$ error (*i.e.*, Euclidean distance between predicted and ground-truth 3D point positions at the horizon) under identical inference settings on the LIBERO evaluation set. RoboFlow4D attains the lowest error (0.0142). In contrast, discarding the context token increases the error to 0.0152, omitting query points increases it to 0.0158, and removing 3D alignment yields the largest error of 0.0160, indicating that each module contributes to accurate 4D prediction.

*Table 3.* **Ablation on Modular Design.** Experiments are conducted in the same inference settings.

| Method | $\ell_2$ Error ↓ |
|---|---|
| **RoboFlow4D** | 0.0142 |
| w/o Context Token | 0.0152 |
| w/o Query Points | 0.0158 |
| w/o 3D Alignment | 0.0160 |

**Dual-System Frequency Ablation.** Table 4 studies the impact of varying update frequency ratios between the fast and slow systems, where the fast policy executes $r$ low-level steps per one slow-system update. As shown in the table, the success rate remains steady across different $r \in \{4, 2, 1\}$ for both DP and DiT controllers, indicating that our slow-fast framework is stable and robust to various update schedules.

*Table 4.* **Ablation on Dual-System Frequency.** We evaluate the robustness of our dual-system on LIBERO-10 and report the average success rate (%). The fast system executes $r$ low-level steps per one slow-system update ($r \in \{4, 2, 1\}$).

| Method | | Success rate (%). |
|---|---|---|
| | Fast/Slow $r=4$ | 71.0 |
| **DP + RoboFlow4D** | Fast/Slow $r=2$ | 71.8 |
| | Fast/Slow $r=1$ | **72.0** |
| | Fast/Slow $r=4$ | 74.6 |
| **DiT + RoboFlow4D** | Fast/Slow $r=2$ | **75.2** |
| | Fast/Slow $r=1$ | 75.0 |

## 4.4. Real-World Experiments

**Robot Platform.** Figure 4 shows the real-world platform, where a 6-dof ROKAE robotic arm equipped with a JODELL gripper is adopted. Visual observations are captured using two RealSense D435 cameras, in wrist-mounted and third-person views. A single NVIDIA RTX 6000 GPU is used for all experiments.

**Baseline.** We compare our method with two open-source state-of-the-art methods for real-world experiments, including $\pi_0$ (Black et al., 2026) and $\pi_0$-Fast (Pertsch et al., 2025).

**Tasks and Datasets.** We validate the generalization of our approach on 4 representative real-world tasks. (1) **Pick-and-Place**: Pick up the cup and place it in the white box. (2) **Stack**: Pick up the red cube and place it on the blue cube. (3) **Assemble**: Pick up the brown cup and insert it into the black cup. (4) **Drawer**: Open the top drawer, place the red cube inside, and close it. For each task, 50 teleoperated demonstrations are collected using a 3D mouse. To introduce mild initial-state diversity, we randomize object poses within a $\pm 2$ cm range during data collection.

**Results Analysis.** Table 5 validates the real-world performance, in terms of success rate and efficiency measured by completion time. As shown in the table, RoboFlow4D

*Table 5.* **Real-world performance in terms of Success rate (%) and efficiency (completion time (seconds))**. Each task is evaluated over an average of 20 trials. **Best** results are in **bold** and second-best results are underlined.

| Method | Pick-and-Place | | Stack | | Assemble | | Drawer | | Avg. | |
|---|---|---|---|---|---|---|---|---|---|---|
| | Succ. ↑ | Time (s) ↓ | Succ. ↑ | Time (s) ↓ | Succ. ↑ | Time (s) ↓ | Succ. ↑ | Time (s) ↓ | Succ. ↑ | Time (s) ↓ |
| $\pi_0$-Fast (Pertsch et al., 2025) | 70.0 | 55.9 | 20.0 | 41.6 | 30.0 | 56.3 | 10.0 | 88.2 | 32.5 | 60.5 |
| $\pi_0$ (Black et al., 2026) | 80.0 | 37.6 | **30.0** | 28.0 | **40.0** | 35.3 | 15.0 | 62.0 | 41.3 | 40.7 |
| DP (Chi et al., 2023) | 60.0 | 31.0 | 20.0 | 26.4 | 25.0 | 34.2 | 5.0 | 61.2 | 27.5 | 38.2 |
| **w/ RoboFlow4D** | 80.0 | **30.0** | 25.0 | **26.2** | 40.0 | 31.0 | 15.0 | **60.0** | 40.0 | **36.8** |
| Δ | **+20.0** | **-1.0** | **+5.0** | **-0.2** | **+15.0** | **-3.2** | **+10.0** | **-1.2** | **+12.5** | **-1.4** |
| DiT (Hou et al., 2024) | 70.0 | 32.5 | 20.0 | 28.1 | 30.0 | 35.1 | 10.0 | 62.2 | 32.5 | 39.5 |
| **w/ RoboFlow4D** | **90.0** | 31.0 | 25.0 | 26.8 | **40.0** | 33.2 | **20.0** | 62.0 | **43.8** | 38.3 |
| Δ | **+20.0** | **-1.5** | **+5.0** | **-1.3** | **+10.0** | **-1.9** | **+10.0** | **-0.2** | **+11.3** | **-1.2** |

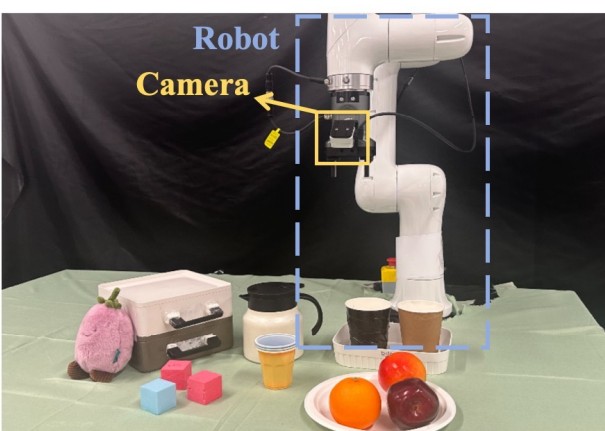

*Figure 4.* Real-world robot platform.

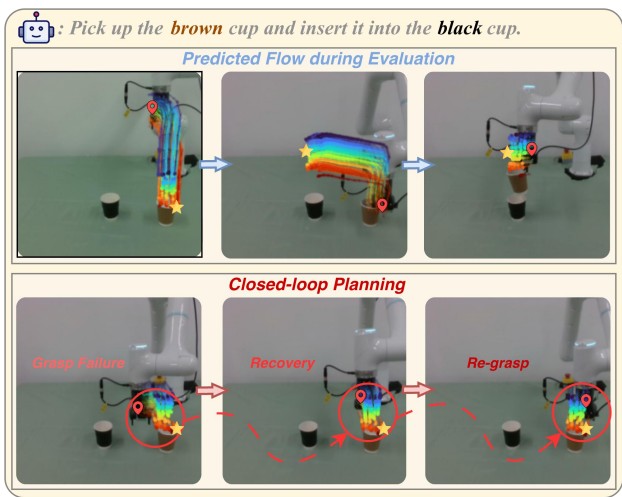

*Figure 5.* **Qualitative Evaluation. Top:** Given a 2D observation and the task instruction, RoboFlow4D predicts goal-oriented flow plans for serial atomic tasks to guide the robot's manipulation until task completion. **Bottom:** In the observation-planning-execution closed loop, when errors occur (*e.g.*, grasp failure), RoboFlow4D correctively re-plans flows to re-align the policy (*recovery*), such that the robot can successfully *re-grasp* the object.

consistently improves success rates by a large margin and reduces task completion time across various policies, including DP and DiT. For example, DP achieves a 12.5% higher average success rate (SR) while reducing completion time (s) by an average of 1.4 s, as evidenced by Pick-and-Place (+20.0 % SR, −1.0 s) and Assemble (+15.0 % SR, −3.2 s) as well. Similar results from DiT can also be observed, such as 43.8% SR, 38.3 s in Avg., notably surpassing the state-of-the-art approach $\pi_0$'s 41.3% and 40.7 s. Generally, both DP and DiT equipped with RoboFlow4D achieve better or competitive success rates and less task completion time compared to other approaches. This can be attributed to the benefits of goal-oriented flow plan guidance for the action policy, which provides more efficient and accurate trajectories that mitigate dithering and unnecessary corrective robot motions caused by real-world uncertainty. Figure 5 qualitatively shows RoboFlow4D's multi-frame 3D flow predictions for sequential atomic tasks and corrective re-planning. Specifically, when a manipulation error occurs (*e.g.*, mis-grasping of the cup), RoboFlow4D re-predicts a corrective flow plan to drive the gripper to recover and complete a successful re-grasping of the cup. This indicates RoboFlow4D's accurate robot motion priors enable the policy to reach the goal more efficiently. Additional qualitative results are provided in Appendix B.

**Latency and Efficiency Discussion.** Beyond execution efficiency, we compare our method with modular pipelines in *flow planning latency*. For example, Dream2Flow (Dharmarajan et al., 2025) reports **3–11 min** to obtain 3D object flow with stacked expert modules, and NovaFlow (Li et al., 2025a) takes ∼**2 min**, which both rely on > 1**B-parameter** video generation experts. In contrast, our lightweight (**0.76B**-parameter) RoboFlow4D directly predicts the *4D motion prior* in a single forward pass within **1 s** without video synthesis, enabling efficient closed-loop manipulation.

## 5. Conclusion

In this paper, we propose RoboFlow4D, a lightweight end-to-end 4D world model that directly predicts goal-oriented multi-frame 3D flows from 2D observations and textual instructions. RoboFlow4D offers a 3D planning interface that avoids costly modular pipelines, introducing a novel paradigm for low-cost 3D flow-conditioned policy learning. By integrating RoboFlow4D with a lightweight action pol-

icy, we develop an observation-planning-execution closed loop for robotic manipulation, where flow planning and action control operate in a slow-fast collaborative manner. Empirical studies in both simulated and real-robot settings demonstrate that RoboFlow4D consistently improves manipulation success rates and efficiency.

## Impact Statement

The proposed RoboFlow4D provides a low-cost 3D flow planning interface for robotic manipulation. It potentially increases manipulation success rates and reduces the cost to deploy reliable robotic systems in diverse industrial and service scenarios. Potential risks include misuse in unsafe automation, biased or brittle behaviors under distribution shifts, and harm caused by prediction errors in dynamic environments. To mitigate these risks, future deployments should incorporate safety constraints, uncertainty-aware planning with fallback behaviors, and rigorous evaluation across diverse scenes before real-world use.

## Acknowledgments

This work is supported in part by Shenzhen Science and Technology Program under grant KJZD20240903104008012, Shenzhen Science and Technology Program under grant ZDCY20250901113000001, CUHK-CUHK(SZ)-GDSTC Joint Collaboration Fund No. 2025A0505000053, GuangDong Key Laboratory of Big Data Computing (2021B1212040002), and the Research Grants Council of the Hong Kong SAR, China (Project No. PolyU 15215824).

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

## A. Latency Analysis

**Video-Generation-based Point Flow Estimation.** Dream2Flow (Dharmarajan et al., 2025) and NovaFlow (Li et al., 2025a) derive 3D object/actionable flow by first generating task-conditioned videos and then applying a multi-stage lifting pipeline (e.g., depth estimation, segmentation, point tracking, and 3D reconstruction). Due to the heavy reliance on video generation, both methods incur *minute-level* end-to-end latency: Dream2Flow reports **3–11 minutes** depending on the chosen video generator, and NovaFlow reports **around 2 minutes** on a single NVIDIA H100 GPU, with runtime dominated by video generation (and subsequent 3D lifting). While such approaches can provide strong zero-shot priors, their high planning latency and dependence on multiple heavyweight modules make fast replanning and real-time deployment challenging. In contrast, RoboFlow4D offers an end-to-end, lightweight alternative that predicts a 4D motion prior directly from observations and language, yielding a more favorable latency–efficiency trade-off for practical robot execution.

**Dual-system Inference Latency.** We utilize one NVIDIA GeForce RTX 6000 consumer-grade GPU. We report wall-clock inference latency measured with batch size 1 on our deployment hardware. RoboFlow4D takes 0.68 s to generate one flow plan. Conditioned on this plan, the low-level action policy takes 0.20 s per forward pass and outputs an action chunk of $H=20$ steps. In our fast–slow design, RoboFlow4D is invoked at a lower frequency to refresh the motion prior, while the action policy can be executed multiple times using the latest predicted flow.

## B. Additional Visualization

**Visualization.** We visualize RoboFlow4D's predicted flows across simulation and real-world benchmarks, including LIBERO (Fig. 6), ManiSkill (Fig. 7), and real-world experiments (Fig. 8).

**Robot–Object Relation.** Within an atomic task stage, diverse intermediate states can correspond to the same stage-level goal. Accordingly, RoboFlow4D learns a goal-conditioned relation between the intended outcome and the motion of the manipulated object, rather than memorizing a single state trajectory. The predicted robot-centric flow is implicitly coupled with object motion, capturing interaction-relevant dynamics. At inference time, while the semantic goal remains fixed, the model updates its flow prediction in a closed-loop manner conditioned on the current state, enabling online progress tracking and corrective adjustments when deviations occur.

## C. Data Generation

**Data Source.** We collect diverse robot data including simulation data and real-world data. Pretraining leverages data from the Droid dataset (Khazatsky et al., 2024). For simulation data, we use the LIBERO and ManiSkill datasets. For real-robot manipulation, we collect task-specific datasets corresponding to each real-world manipulation task.

**Robot Gripper Grounding and Flow Tracking.** To capture embodiment motion, we construct *gripper-centric* flows from third-person RGB observations. We first apply Grounded-SAM2 to segment the robot gripper and obtain a binary mask. Conditioned on this mask, we use SpatialTrackerV2 (Xiao et al., 2025) to track 3D point flows on the gripper throughout each episode. Since raw point trajectories can contain redundant or noisy signals, we apply a three-stage filtering pipeline: (i) remove near-static tracks, (ii) reject outlier points, and (iii) discard tracks with implausibly large inter-frame displacements. For datasets without a visible gripper, we instead use scene-level flow as a fallback.

**Atomic Task Decomposition.** Predicting long-horizon flows for an entire task is often challenging and brittle. We consider a long-horizon manipulation task $\mathcal{T}$ specified by a language instruction $\ell$ and an initial observation $\mathbf{o}_0$, together with its corresponding demonstration $\tau = \{(\mathbf{o}_t, \mathbf{a}_t)\}_{t=0}^{H-1}$, where $\mathbf{o}_t$ is the image observation and $\mathbf{a}_t$ is the robot action at time $t$.

A long-horizon manipulation task typically consists of a sequence of short, physically meaningful interaction phases (*e.g.*, pre-grasp $\rightarrow$ grasp $\rightarrow$ transport $\rightarrow$ release), motivating decomposition into atomic tasks for stable hierarchical imitation learning (Sutton et al., 1999; Argall et al., 2009; Kipf et al., 2019; Aytar et al., 2018). Formally, we partition $\tau$ into $M$ contiguous segments $\{\tau_i\}_{i=1}^{M}$ with boundaries $0 = s_1 \leq e_1 < s_2 \leq \cdots < s_M \leq e_M = H-1$, where $\tau_i = \{(\mathbf{o}_t, \mathbf{a}_t)\}_{t=s_i}^{e_i}$. We treat the terminal observation $\mathbf{o}_{e_i}$ as the **atomic goal** for segment $\tau_i$.

We use changes in the gripper state as the cue for atomic task decomposition. Let $g_t$ denote the gripper open/close signal, and define its binarized state $b_t = \mathbb{I}[g_t > 0] \in \{0, 1\}$. We define an atomic task with a maximal contiguous interval $[s_i, e_i]$ such that $b_t$ remains constant for all $t \in [s_i, e_i]$, and a boundary occurs when the gripper state changes (*i.e.*, $b_{e_i} \neq b_{e_i+1}$). Intuitively, open$\leftrightarrow$close transitions correspond to salient physical interaction events (*i.e.*, grasp/release), producing segment

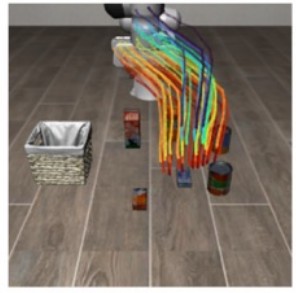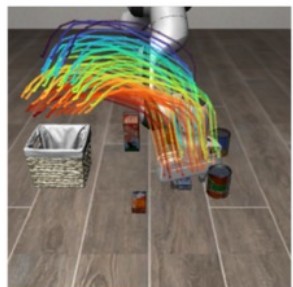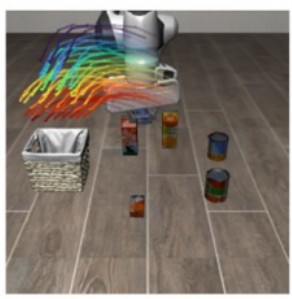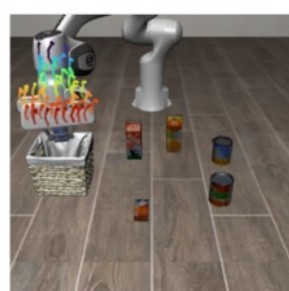

*Pick up the cream cheese and place it in the basket*

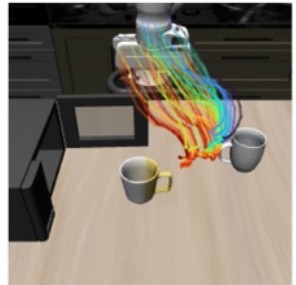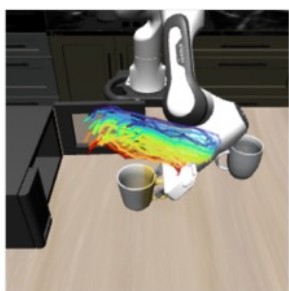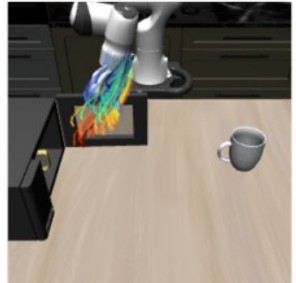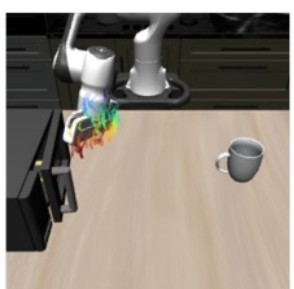

*Put the yellow and white mug in the microwave and close it*

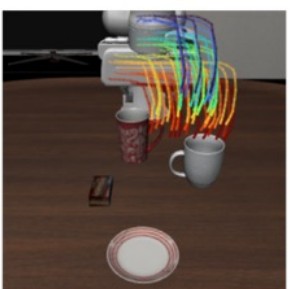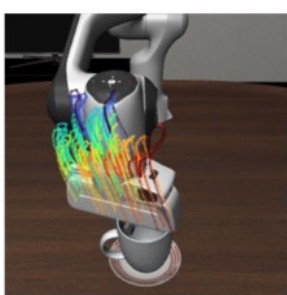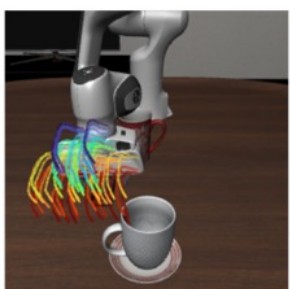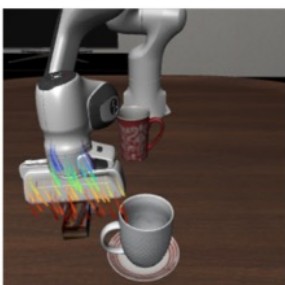

*Put the white mug on the plate and put the chocolate pudding to the right of the plate*

*Figure 6.* LIBERO Visualization.

boundaries. In practice, to suppress spurious gripper oscillations, we only accept a boundary if the new state persists for a short temporal window, and we discard segments shorter than $\ell_{\min}$. If no stable transition is found, we keep the trajectory as a single segment.

**Goal-Oriented Point Flow Resampling.** For each segment $\tau_i = [s_i, e_i]$, we construct a compact supervision that emphasizes key interactions near the atomic goal. Let $\mathbf{P}_t \in \mathbb{R}^{N \times 3}$ denote the 3D positions of $N$ tracked query points at time $t$. Rather than supervising at every timestep, we resample $K$ key frames $\{t_k\}_{k=0}^{K-1}$ within $[s_i, e_i]$ using a monotone time-warping rule that densifies samples close to the goal:

$$u_k = \left(\frac{k}{K-1}\right)^{\gamma}, \quad t_k = \lfloor s_i + u_k \left(e_i - s_i\right)\rfloor, \tag{7}$$

where $\gamma > 1$ allocates more samples near $e_i$ (the atomic goal). We then collect point flows at these resampled frames:

$$\mathbf{x}_0[k, n] = \mathbf{P}_{t_k}[n], \qquad \mathbf{x}_0 \in \mathbb{R}^{K \times N \times 3}. \tag{8}$$

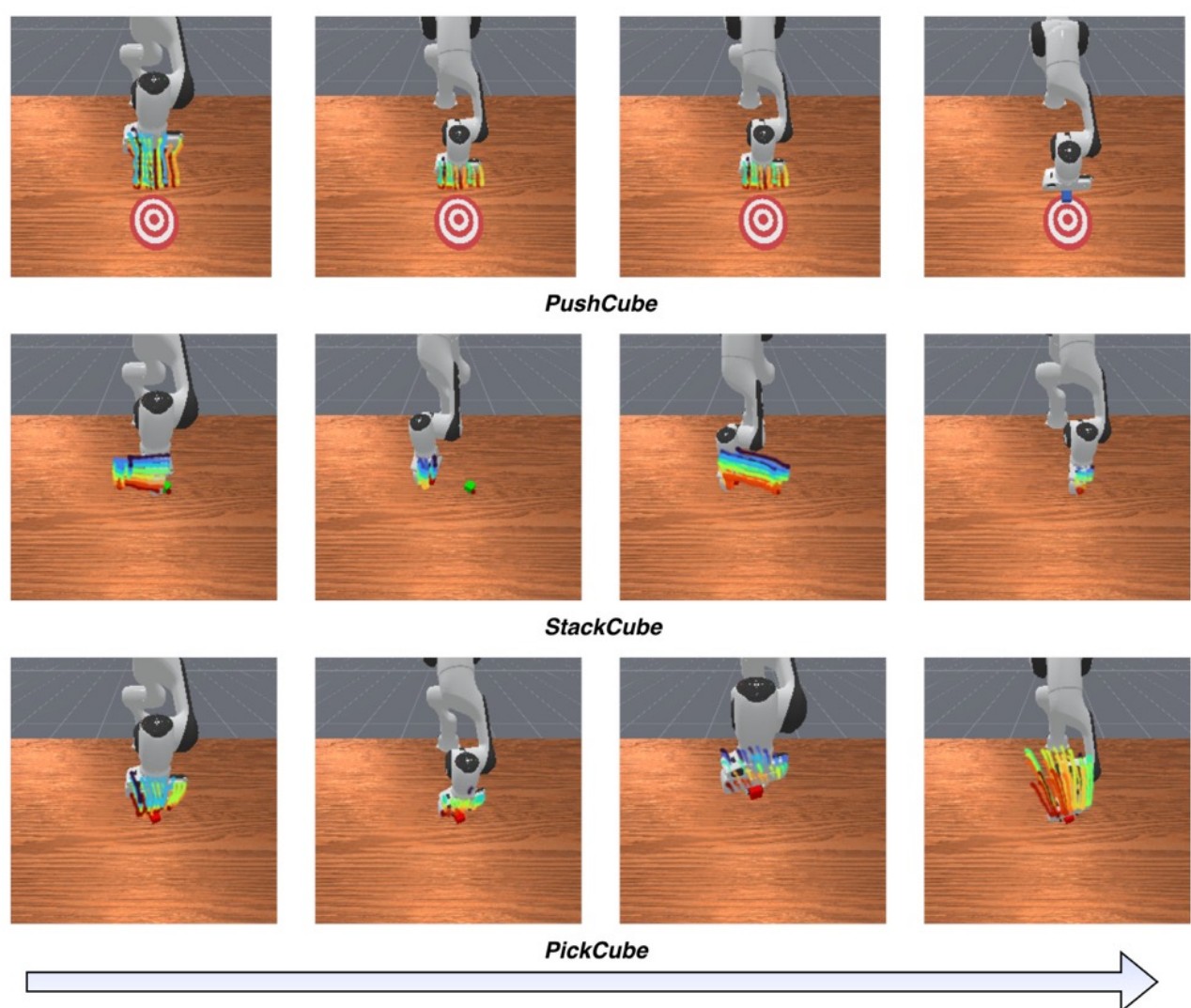

*Figure 7.* Maniskill Visualization.

## D. Training Details

We train our model via conditional denoising (Ho et al., 2020). Let $\mathbf{x}_0 \in \mathbb{R}^{K \times N \times 3}$ be ground-truth flow trajectories. We sample a timestep $t \sim \mathcal{U}\{1, \ldots, T\}$ and corrupt $\mathbf{x}_0$ through the forward process:

$$\mathbf{x}_t = \sqrt{\bar{\alpha}_t}\, \mathbf{x}_0 + \sqrt{1 - \bar{\alpha}_t}\, \boldsymbol{\epsilon}, \quad \boldsymbol{\epsilon} \sim \mathcal{N}(\mathbf{0}, \mathbf{I}), \tag{9}$$

where $\bar{\alpha}_t$ is the cumulative product of noise schedule coefficients. We use the stable $v$-prediction parameterization (Salimans & Ho, 2022):

$$\mathbf{v}_t = \sqrt{\bar{\alpha}_t}\, \boldsymbol{\epsilon} - \sqrt{1 - \bar{\alpha}_t}\, \mathbf{x}_0, \tag{10}$$

and train the network $\mathbf{v}_\theta(\mathbf{x}_t, t, \mathbf{c}, \mathbf{m})$ to predict $\mathbf{v}_t$, where $\mathbf{c}$ and $\mathbf{m}$ denote condition and context.

In addition, we adopt classifier-free conditioning dropout (Ho & Salimans, 2022). During training, we drop the condition $\mathbf{c}$ with probability $p_{\text{uncond}}$, enabling the network $\mathbf{v}_\theta(\mathbf{x_t}, \mathbf{t}, \mathbf{c}, \mathbf{m})$ to handle both conditional and unconditional inputs. At inference time, we apply classifier-free guidance:

$$\hat{\mathbf{v}}_\theta = (1 + w)\, \mathbf{v}_\theta(x_t, t, c) - w\, \mathbf{v}_\theta(x_t, t, \varnothing), \tag{11}$$

where $w$ is the guidance scale and $\varnothing$ denotes the dropped condition.

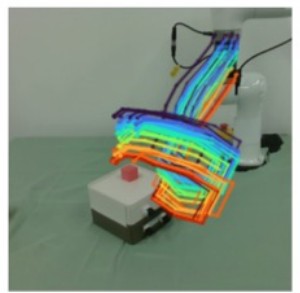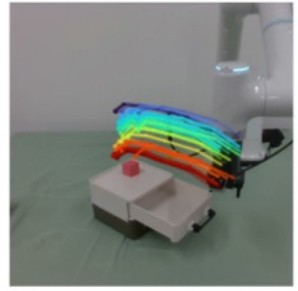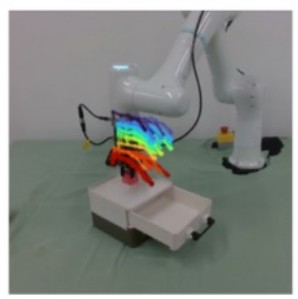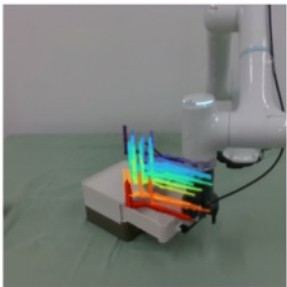

*Put the red cube into the closed top drawer and close the drawer*

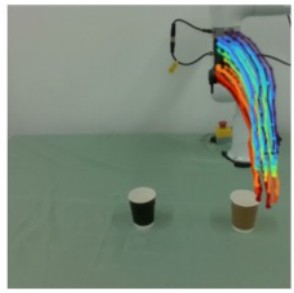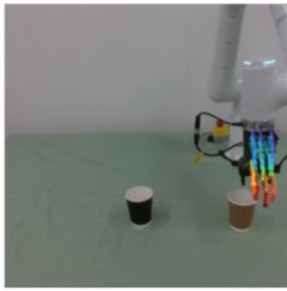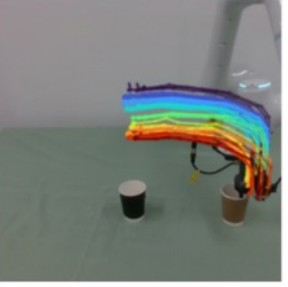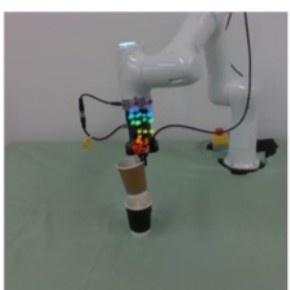

*Pick up the brown cup and place it inside the black cup*

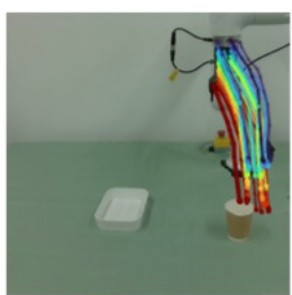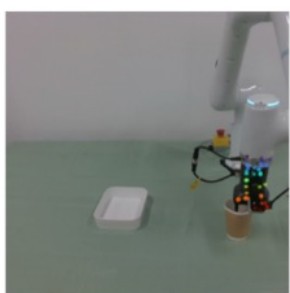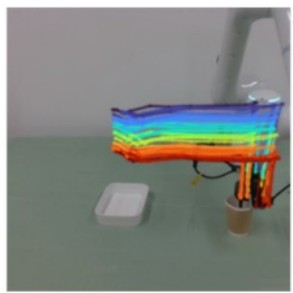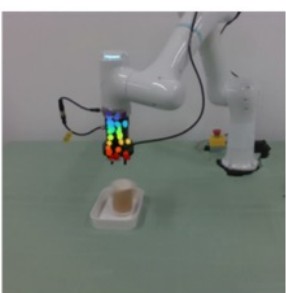

*Pick up the cup and place it inside the white box*

*Figure 8.* Real-world Visualization.

**Diffusion Loss.** It computes a visibility-weighted mean-squared error:

$$\mathcal{L}_{\text{diff}} = \mathbb{E}_{t,\epsilon}\left[\frac{1}{\sum_{k,n} w_{k,n}} \sum_{k=1}^{K} \sum_{n=1}^{N} w_{k,n} \left\|\mathbf{v}_\theta(\mathbf{x}_t, t, \tilde{\mathbf{c}}, \mathbf{m})[k,n] - \mathbf{v}_t[k,n]\right\|_2^2\right], \tag{12}$$

where $w_{k,n} \in [0,1]$ reduces the weight of occluded or low-confidence points.

**Alignment Loss.** RoboFlow4D takes as input 2D observations and language, while predicting multi-frame 3D future motion. This dimension lifting is inherently ambiguous. To alleviate this issue, we introduce a 3D alignment loss $\mathcal{L}_{\text{align}}$. Specifically, we distill the pooled 3D conditioning feature $\mathbf{c}_{3\text{D}}$ toward a teacher embedding $\mathbf{h}$ extracted by a frozen VGGT model (Wang et al., 2025). A learnable projector $g_\phi(\cdot)$ maps $\mathbf{c}_{3\text{D}}$ to the teacher space.

$$\mathcal{L}_{\text{align}} = \left\|g_\phi(\mathbf{c}_{3\text{D}}) - \mathbf{h}\right\|_2^2. \tag{13}$$

**Smoothness Loss.** To improve the temporal consistency of generated point flows, we regularize the denoised flows $\hat{\mathbf{x}}_0$ by penalizing second-order temporal differences:

$$\Delta^2 \hat{\mathbf{x}}_0[k,n] = \hat{\mathbf{x}}_0[k+1,n] - 2\hat{\mathbf{x}}_0[k,n] + \hat{\mathbf{x}}_0[k-1,n], \tag{14}$$

with a robust Charbonnier penalty (Barron, 2019):

$$\mathcal{L}_{\text{smooth}} = \frac{1}{\sum_{k,n} w_{k,n}} \sum_{k=2}^{K-1} \sum_{n=1}^{N} w_{k,n} \sqrt{\|\Delta^2 \hat{\mathbf{x}}_0[k,n]\|_2^2 + \epsilon^2}. \tag{15}$$

Therefore, the **overall training objective** is

$$\mathcal{L} = \mathcal{L}_{\text{diff}} + \lambda_{\text{align}} \mathcal{L}_{\text{align}} + \lambda_{\text{smooth}} \mathcal{L}_{\text{smooth}}, \tag{16}$$

where $\lambda_{\text{align}}$ and $\lambda_{\text{smooth}}$ are scalar weighting coefficients.

**Training Cost.** RoboFlow4D is pretrained using 8 A800 GPUs for one week. It can be further fine-tuned using 8 A800 GPUs for an additional day.

## E. Calibrating 3D Flow for Motion Planning

To enable motion planning, we express RoboFlow4D's predicted 3D flow in a metric, robot-centric coordinate frame. Since the raw flow may inherit scale ambiguity from the underlying 3D reconstruction model, we use RGB-D observations with camera intrinsics and camera-to-robot extrinsics to lift image points into metric 3D space. We then estimate a robust global alignment between the predicted trajectories and the RGB-D lifted trajectories, yielding temporally coherent, metric-scale 3D displacements that can serve as motion priors for planning and model-based control.

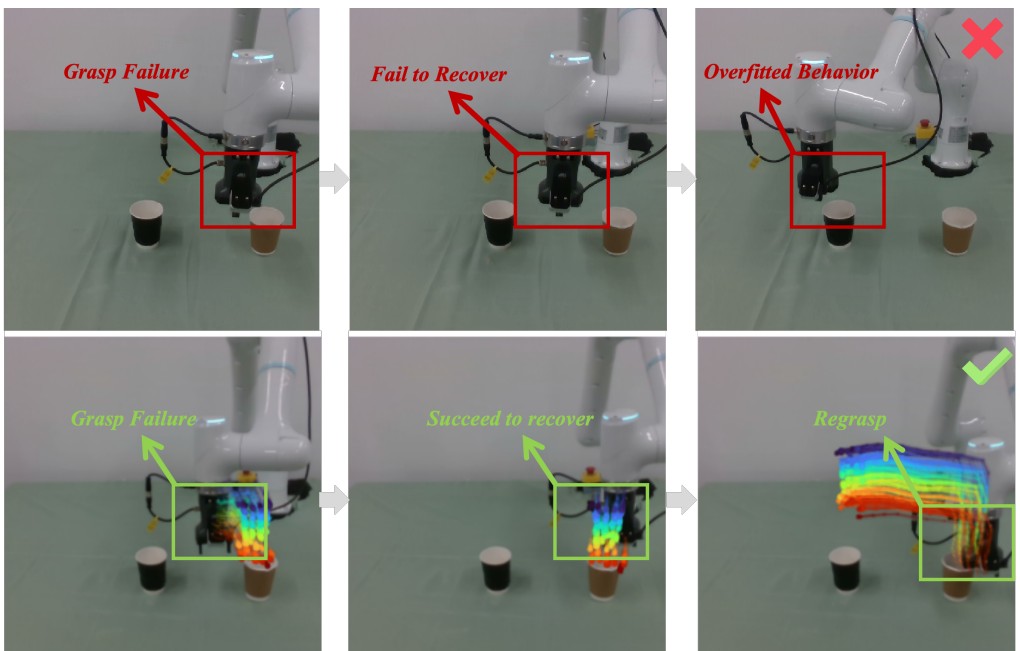

*Figure 9.* Recovery comparison after failure. RoboFlow4D replans with flow guidance and successfully recovers from grasp failure.

## F. Failure-Mode Analysis and Design Choices

We analyze representative failure modes of existing flow-based manipulation methods and explain how these observations motivate the design of RoboFlow4D. Beyond the fundamental limitation that 2D flow lacks explicit 3D geometry, Figs. 9, 10, and 11 further illustrate three practical failure modes: overly long-horizon flow prediction, limited recovery ability, and error accumulation in modular video-based pipelines.

**3D Flow over 2D Flow.** 2D flow-based methods (Xu et al., 2024) can provide intuitive motion cues in the image plane, but their predictions are defined in pixel space and do not explicitly encode depth, object geometry, or physical spatial constraints.

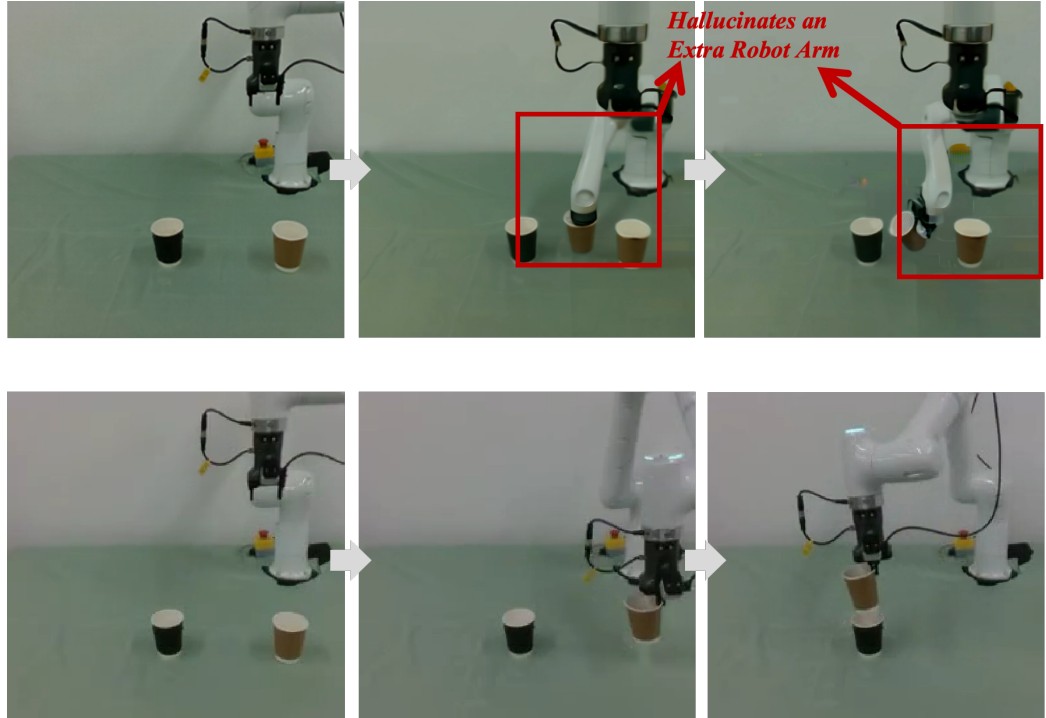

*Figure 10.* Failure of modular video-based flow prediction. Video generation may hallucinate implausible robot motions, corrupting subsequent tracking and flow estimation.

As a result, a visually plausible 2D trajectory may still correspond to an infeasible or ambiguous 3D motion for the robot. RoboFlow4D instead predicts 3D flows, which provide geometry-aware motion priors for downstream manipulation and better align the predicted motion with the physical workspace.

**Goal-Oriented Adaptive Flow Planning.** Fixed-horizon flow prediction can be difficult to execute when the predicted trajectory spans multiple manipulation phases. As shown in Fig. 11, the predicted flow in the upper row covers two atomic tasks at once, causing the robot to execute an ambiguous and physically challenging motion. This often leads to manipulation failure, especially in long-horizon tasks that involve sequential subgoals such as approaching, grasping, lifting, and placing. In contrast, RoboFlow4D predicts flow toward the current atomic goal. This goal-oriented design provides a more localized and executable motion prior, allowing the robot to complete each manipulation stage more reliably.

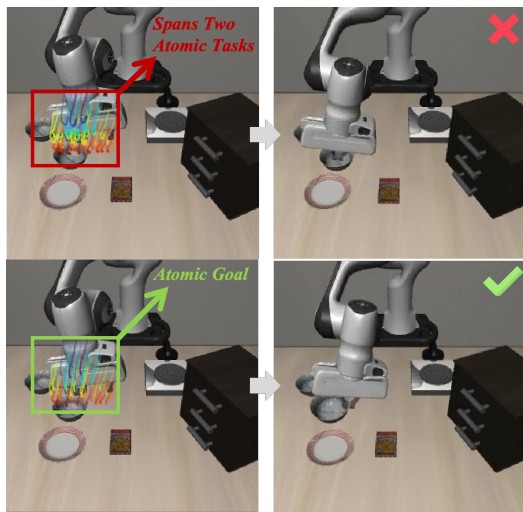

*Figure 11.* Failure of long-horizon flow prediction across multiple atomic tasks. Fixed-horizon flow may span two atomic goals and cause execution failure, while RoboFlow4D predicts flow toward the current atomic goal.

**Recovery-Aware Replanning.** Manipulation policies often fail when the execution deviates from the expected state. For example, as illustrated in Fig. 9, the baseline policy first suffers from a grasp failure, then fails to recover, and finally falls into overfitted behavior that no longer contributes to task completion. RoboFlow4D mitigates this issue by using updated flow guidance during execution. When the robot fails to grasp the object, the model can replan according to the current observation, recover from the failed state, and perform regrasping. This

adaptive replanning ability improves robustness under execution errors and intermediate-state deviations.

**End-to-End Flow Prediction.** Some existing 3D flow methods rely on modular pipelines that first generate future videos and then perform point tracking or 3D flow estimation. However, such pipelines are vulnerable to error accumulation. As shown in Fig. 10, the generated video may hallucinate physically implausible content, such as an extra robot arm. Once this unrealistic video is used for tracking, the resulting flow becomes unreliable and can mislead the downstream policy. To avoid this failure mode, RoboFlow4D adopts a lightweight end-to-end framework that directly predicts task-relevant 3D flow, reducing both computational overhead and cascading errors from intermediate video generation.

Overall, these failure cases motivate four key design choices of RoboFlow4D: **3D spatial awareness**, **goal-oriented adaptive planning**, **recovery-aware replanning**, and **lightweight end-to-end 3D flow prediction**. Together, these designs improve the physical feasibility, executability, robustness, and efficiency of flow-guided robotic manipulation.

