# OpenReview forum: "RoboFlow4D: A Lightweight Flow World Model Toward Real-Time Flow-Guided Robotic Manipulation"
_ICML.cc/2026/Conference — ICML 2026 regular_

### Official Review · Reviewer_zmr4 · 2026-02-28

**Soundness:** 3
**Presentation:** 3
**Significance:** 3
**Originality:** 3
**Overall Recommendation:** 5
**Confidence:** 4

**Summary:**

RoboFlow4D is a lightweight, end-to-end 3D world model that enables robots to plan and act efficiently by directly predicting multi-frame 3D motion flows from visual and textual inputs. Unlike traditional modular pipelines that stack multiple heavy models (like depth estimation and point tracking), this unified framework achieves a 120x speedup and reduces model size by 24%, allowing for real-time, low-latency manipulation. By using a "slow-fast" architecture—where a deliberate planner sets the 3D path and a fast controller executes the actions—it improves task success rates by up to 20% in real-world settings while keeping planning time under one second.

**Compliance With Llm Reviewing Policy:**

Affirmed.

**Key Questions For Authors:**

- In real-world experiments, the success rate and efficiency improvements over state-of-the-art baselines like pi0 and Diffusion Policy (DP) appear limited. Could the authors clarify why the explicit 3D flow planning does not yield a more significant advantage in these settings?
- How sensitive is the success rate to the update frequency of the "slow" planning system? If the planning frequency were increased to match the control frequency, would the performance improve, or would the computational overhead negate the benefits?

**Limitations:**

yes

**Strengths And Weaknesses:**

## Strengths
- Unlike previous modular pipelines that stack multiple heavy expert models, RoboFlow4D achieves a 120× speedup and a 24% reduction in model scale, making it highly practical for real-time robotic deployment.
- The claims are well-supported by a diverse set of experiments across both simulation (LIBERO/ManiSkill3) and real-world robotic tasks, demonstrating consistent performance gains over state-of-the-art baselines.
- The presentation is exceptionally clear, well written, and easy to follow.

## Weaknesses
- The dual-system, closed-loop control is a standard paradigm in robotics; the paper does not sufficiently distinguish its "slow-fast" approach from existing hierarchical or asynchronous frameworks.
- In real-world experiments, the model fails to show a significant efficiency or success rate advantage over state-of-the-art baselines like pi0 and DP, making the practical benefits of the unified flow model less convincing.

---

> ### Author Rebuttal · Authors · 2026-03-31
>
> Thank you very much for your in-depth review and kind approval (*e.g.*, ***highly practical value, strong experimental support, and clear presentation***). Below, we provide detailed responses to each of your constructive comments.
>
> ---
>
> **W1. The paper does not sufficiently distinguish its "slow-fast" approach from existing hierarchical or asynchronous frameworks.**
>
> Thanks for pointing this out. Our slow-fast framework differs from existing hierarchical or asynchronous frameworks in three key aspects.
>
> 1. **Planning Objective**: According to the Slow System's planning objective, dual-system frameworks can be broadly categorized into three paradigms: instruction-based [1][2], latent variable-based [3], and flow-based (ours). Unlike textual instructions and latent variables, our Slow System produces explicit 3D flows, providing a strong motion prior for the Fast System's action generation.
> 2. **Lightweight Design**: Instead of relying on a heavy VLM as the Slow System [1][2][3], we develop a lightweight, end-to-end flow world model as the Slow System, improving overall efficiency.
> 3. **Closed-loop Flow-guided Control**: Unlike the existing flow-guided hierarchical framework [4], which is open-loop, ours enables closed-loop interaction between the flow planner and action executor throughout the manipulation process.
>
> In response to this valuable feedback, we will revise the paper to highlight these distinctions.
>
>
> References:
>
> [1] X. Shi, B. Ichter, M. Equi, et al. Hi robot: Open-ended instruction following with hierarchical vision-language-action models. ICML 2025.
>
> [2] T. Jiang, T. Yuan, Y. Liu, et al. Galaxea Open-World Dataset and G0 Dual-System VLA Model. arXiv:2509.00576, 2025.
>
> [3] H. Chen, J. Liu, C. Gu, et al. Fast-in-Slow: A Dual-System VLA Model Unifying Fast Manipulation within Slow Reasoning. NeurIPS 2025.
>
> [4] H. Li, L. Sun, Y. Hu, et al. Novaflow: Zero-shot manipulation via actionable flow from generated videos. arXiv:2510.08568, 2025.
>
> ---
>
> **W2 \ Q1. Why does explicit 3D flow planning yield limited real-world gains over strong baselines such as pi0 and DP?**
>
> Many thanks for this insightful question. We would like to clarify that the practical benefit of explicit 3D flow planning is not to create an overwhelmingly large margin over already strong VLA models such as pi0, but to provide a lightweight, low-latency motion prior that can consistently improve downstream control.
>
> 1. **Significant improvements over DP/DiT**. In the real-world setting, comparing the same baseline policies with and without RoboFlow4D (Table 5 of the paper):  DP improves from 27.5% to 40.0% in average success rate, and DiT improves from 32.5% to 43.8%. Besides, both also reduce average completion time. These consistent gains demonstrate the value of 3D flow planning, which provides more accurate and efficient 3D trajectories for manipulation.
> 2. **Competitive or superior to pi0**. Both DP and DiT equipped with RoboFlow4D match or surpass pi0 (a larger VLA model) in success rate (40.0%/43.8% vs. 41.3%) and completion time (36.8 s/38.3 s vs. 40.7 s). This is notable given that our lightweight framework uses only about 1/3 of pi0's parameters. Besides, explicit 3D flow planning as an additional input can also boost pi0 on the assemble task, as shown in Table 1.
>
> **Table 1. Real-world performance on the Assemble task.**
> |Method | Success Rate (%) $\uparrow$ | Completion Time (s) $\downarrow$ |
> |-|-|-|
> | pi0  w/o RoboFlow4D| 40.0 | 35.3 |
> | pi0  w RoboFlow4D  | 50.0 | 34.8 |
>
>  ---
>
> **Q2. How sensitive is the success rate to the update frequency of the "slow" planning system?**
>
> Thanks for this great question. We conducted a frequency ablation (Table 2), varying the ratio between control and planning frequency from 1 to 8. The results show that our method is relatively insensitive to the planner update frequency within a specific range.
>
> Specifically, the success rate remains stable (71.0%, 71.8%, 72.0%) across frequency ratio `R ∈ {4, 2, 1}`, indicating that a single goal-oriented flow plan provides sufficient guidance for multiple action chunks. When `R > 4`, performance drops obviously. Therefore, we recommend setting the planner update frequency to no less than 1/4 of the control frequency. In our work, we set `R = 2` to achieve a good trade-off between performance and computational overhead.
>
> **Table 2. Ablation on Dual-System Frequency.**
> | Method | Frequency Ratio | Success Rate (%) $\uparrow$ |
> |-|-|-|
> | DP + RoboFlow4D | Fast/Slow \(R=8\) | 67.2 |
> | DP + RoboFlow4D | Fast/Slow \(R=4\) | 71.0 |
> | DP + RoboFlow4D | Fast/Slow \(R=2\) | 71.8 |
> | DP + RoboFlow4D | Fast/Slow \(R=1\) | 72.0 |

---

> > ### Author Rebuttal · Reviewer_zmr4 · 2026-04-03
> >
> > Appreciate the author’s response. My questions have been successfully resolved.

---

> > > ### Author Response · Authors · 2026-04-03
> > >
> > > We are glad to have addressed your concerns. Thank you for your thoughtful feedback and expertise throughout the review process.

---

### Official Review · Reviewer_1qSG · 2026-03-11

**Soundness:** 3
**Presentation:** 3
**Significance:** 3
**Originality:** 3
**Overall Recommendation:** 5
**Confidence:** 4

**Summary:**

This paper proposes RoboFlow4D, a lightweight end-to-end world model that directly predicts goal-oriented multi-frame 3D flow from historical RGB observations, optional 2D query points, and text instructions, and uses this prediction as an explicit motion prior for downstream action policies. At the system level, the authors further introduce a slow-fast closed-loop control framework that combines low-frequency flow planning with high-frequency action execution. The framework is evaluated in simulation on LIBERO and ManiSkill3, as well as on four real-world robotic tasks.

**Compliance With Llm Reviewing Policy:**

Affirmed.

**Final Justification:**

Considering both the original submission and the rebuttal, my final score for this paper is **5 (Accept)**.

First, in terms of **Soundness** and **Originality**, the paper identifies a meaningful issue in current 3D flow methods, namely that they often rely on heavyweight module stacking and are therefore difficult to deploy in real time. I find this to be a well-motivated and worthwhile research problem, and the paper does a good job of making the motivation clear.

Second, the rebuttal strengthens the empirical case by adding comparisons with recent 3D flow planning methods. Together with the comparisons to VLA-style methods already included in the main paper, this results in a sufficiently strong baseline set, which increases my confidence in the proposed model. Combined with the rich qualitative visualizations, I believe the paper reaches a **good** level in terms of **Presentation** and **Significance** as well.

That said, although the paper is strong overall — from motivation to method to experimental validation — I am not giving it a top score. The main reason is that there has been a large amount of recent work around flow-guided methods over the past two years, so in my view the level of novelty does not quite reach the level of **excellent**. Still, this does not affect my overall view that the paper is of high quality and merits acceptance.

**Key Questions For Authors:**

1. Why is there no direct comparison with the most relevant 3D flow planning methods?

2. Please provide an independent ablation specifically for the adaptive temporal horizon itself.

3. Please report statistical robustness: the real-world evaluation appears to use only around 20 trials per task, and the simulation tables also report only point estimates of success rate. Please provide results across different random seeds, multi-run training variance, confidence intervals, or at least significance analysis for the key improvements.

**Limitations:**

yes

**Strengths And Weaknesses:**

Strengths
1. The problem setting is important, and the paper has a clear objective.
The paper addresses a meaningful problem: 2D flow lacks geometric feasibility, while existing 3D flow methods often rely on heavyweight module stacking and are therefore difficult to deploy in real time. The authors clearly formulate their goal as “end-to-end, lightweight, explicit 3D flow planning with closed-loop control.” This objective is consistent throughout both the method and the experiments, and the motivation is clearly articulated.

2. The experiments cover both simulation and the real world, and the authors attempt to validate efficiency.
The paper demonstrates consistent gains over DP/DiT across LIBERO, ManiSkill3, and real-world robotic tasks. The results show that these improvements persist across multiple tasks and benchmarks, supporting the central empirical claim that explicit 3D flow priors can indeed help action generation. The appendix reports a planning latency of 0.68s, a single forward-pass time of 0.20s for the action policy, and minute-level comparisons with Dream2Flow and NovaFlow, indicating that the authors recognize efficiency as an indispensable part of the paper’s claim rather than focusing only on success rate.

Weaknesses
1. Insufficient direct comparison with the most relevant 3D flow planning methods.
In Related Work (Section 2.2), the paper discusses methods such as PointWorld as the most closely related approaches in 3D trajectory / 3D flow action modeling. However, the actual experimental baselines mainly consist of VLA, BC/ACT, and DP/DiT without flow, rather than controlled comparisons with these most relevant 3D flow planning methods. The current evidence more strongly supports the conclusion that RoboFlow4D improves baseline policies, but is still insufficient to fully establish that it outperforms existing nearest-neighbor 3D flow planners. In addition, the authors do not explain why they do not use RGB-D data, which could potentially provide a stronger geometric prior.

2. The ablation study is not sufficiently comprehensive.
The current ablations mainly examine context tokens, query points, 3D alignment, and the fast/slow frequency ratio, but they do not provide a direct validation that an adaptive temporal horizon is better than a fixed horizon. As it stands, the ablation study is more about showing that certain modules are useful, rather than rigorously validating the paper’s core design claims.

3. Statistical significance and variance analysis are relatively sparse.
Most results report only overall averages. Reporting confidence intervals or standard deviations would provide stronger empirical support.

---

> ### Author Rebuttal · Authors · 2026-03-31
>
> Many thanks for your positive recognition (*e.g.*, **clear and meaningful motivation, extensive experiments, and attention to efficiency**). We appreciate the opportunity to address your valuable feedback as follows.
>
> ---
> **W1-(1)/Q1. Insufficient direct comparison with the most relevant 3D flow planning methods.**
>
> Thanks for raising this concern. RoboFlow4D (ours), Dream2Flow [1], PointWorld [2], and NovaFlow [3] were developed concurrently for 3D flow planning in robotics. We compare with Dream2Flow and NovaFlow below, since PointWorld has not yet released its precomputed datasets and checkpoints.
>
> As shown in Table 1, RoboFlow4D outperforms Dream2Flow and NovaFlow in both effectiveness (*i.e.*, lower flow error and higher success rate) and efficiency (*i.e.*, lower latency, memory, and FLOPs). We attribute these gains to (i) our end-to-end pipeline design, avoiding the error accumulation inherent in Dream2Flow/NovaFlow's modular pipeline; and (ii) our lightweight  architecture design.
>
>
> **Table 1. Quantitative comparison in effectiveness and efficiency.**
> | Method | 3D Flow Error ↓ | End-to-End Latency (s) ↓ | Total GPU Memory (GB) ↓ | Total FLOPs ↓ | Success Rate (%) ↑ |
> |---|---:|---:|---:|---:|---:|
> | Dream2Flow [1] | 0.0354 | 509.35 | 38.18 × 4 | 35.56 PFLOPs | 69.2 |
> | NovaFlow [3] | 0.0719 | 596.20 | 35.33 × 4 | 35.22 PFLOPs | 68.4 |
> | **RoboFlow4D (ours)** | **0.0142** | **0.73** | **3.12** | **14.54 TFLOPs** | **72.0** |
>
> References:
>
> [1] Karthik Dharmarajan, Wenlong Huang, Jiajun Wu, Fei-Fei Li, and Ruohan Zhang. Dream2flow: Bridging video generation and open-world manipulation with 3d object flow. arXiv preprint arXiv:2512.24766, 31/12/2025.
>
> [2] Wenlong Huang, Yu-Wei Chao, Arsalan Mousavian, Ming-Yu Liu, Dieter Fox, Kaichun Mo, Fei-Fei Li. Pointworld: Scaling 3d world models for in-the-wild robotic manipulation. arXiv preprint arXiv:2601.03782, 07/01/2026.
>
> [3] Hongyu Li, Lingfeng Sun, Yafei Hu, Duy Ta, Jennifer Barry, George Konidaris, Jiahui Fu. Novaflow: Zero-shot manipulation via actionable flow from generated videos. arXiv:2510.08568, 09/10/2025.
>
> ---
> **W1-(2). Lack explanation for not using RGB-D.**
>
> Our work focuses on RGB-only, closed-loop manipulation, as this setting is *lower in cost* and *easier to deploy* for robotic tasks. We totally agree that RGB-D may provide a strong geometric prior, and we consider extending RoboFlow4D to depth-enhanced settings as an important future direction.
>
>
> ---
>
> **W2/Q2. The ablation study is not sufficiently comprehensive, specifically for the adaptive temporal horizon.**
>
> Thank you very much for pointing this out. To validate the design choice of an adaptive temporal horizon rather than a fixed horizon, we conduct an ablation study (shown in Table 2) and will further add it into the paper.
>
> **Table 2. Ablation of modules and design choises on LIBERO-10.**
> | Method| Success Rate (%) ↑ |
> |-|-|
> | RoboFlow4D + Diffusion Policy |    **72.0**  |
> | w/o Context Token |   70.8  |
> | w/o Query Points  |   70.4  |
> | w/o 3D Alignment  |   70.2  |
> | with Fixed Temporal Horizon  |   69.4  |
>
>
> Table 2 validates the importance of the adaptive horizon, achieving a higher manipulation success rate than the fixed horizon setting. The advantage stems from two aspects:
> (1) The fixed horizon provides short, local plans whose contribution to overall task completion is often unclear. In contrast, the adaptive horizon generates longer, global plans that provide an explicit flow toward task completion.
> (2) The adaptive horizon naturally adopts a coarse-to-fine strategy. Specifically, it provides coarse trajectory direction guidance in the early stage of the task, and offers fine-grained operation guidance when approaching the target object.
>
> ---
>
>
> **W3/Q3. Statistical significance and variance analysis are relatively sparse. Please report statistical robustness.**
>
> Thanks for this valuable suggestion. We have conducted additional statistical analyses, detailed in Tables A, B, and C at the anonymous link (https://anonymous.4open.science/w/RoboFlow4D/).
>
> The analyses focus on three aspects:
> 1. **Real-world experiments with confidence intervals (Table A)**. We report 95% Wilson score confidence intervals for success rates and mean ± standard deviation for completion times. RoboFlow4D consistently improves base policies, with better confidence intervals (*e.g.*, DiT on Pick-and-Place: 48.1–85.5% → 69.9–97.2%).
> 2. **Manipulation with more random seeds (Table B)**. We evaluate standard deviations for ManiSkill3 manipulation across 100 trials per task with 3 random seeds. The improvements from RoboFlow4D consistently exceed the variance (*e.g.*, DP average: 12.3 ± 1.8% → 22.0 ± 2.4%, a minimum gain of +5.5%).
> 3. **Extended real-world trials (Table C)**. To address the concern about limited trial counts, we conduct additional experiments with 40 trials per task for our methods. The results closely match the original 20-trial evaluations (in Table A).

---

> > ### Author Rebuttal · Reviewer_1qSG · 2026-04-03
> >
> > I thank the authors for their detailed rebuttal. The concerns I previously raised have now been adequately addressed. The additional evidence is helpful and increases my confidence in the empirical support of the paper. In particular, the new ablations and statistical analyses directly answer the questions I raised, and the comparison to closely related concurrent work is valuable within the scope of the rebuttal. Therefore, I will raise my score from Weak Accept to Accept.

---

> > > ### Author Response · Authors · 2026-04-03
> > >
> > > Thank you for your positive feedback. We are glad that our response addressed your concerns. We sincerely appreciate your time, expertise, and recognition of our work.

---

### Official Review · Reviewer_nrX8 · 2026-03-12

**Soundness:** 3
**Presentation:** 3
**Significance:** 3
**Originality:** 3
**Overall Recommendation:** 5
**Confidence:** 3

**Summary:**

This paper proposes RoboFlow4D, which is an end-to-end 3D point track predictor with multiview images as input. RoboFlow4D adopts a slow-fast collaboration for flow and action prediction, and enable high efficiency during inference. Experiments show that proposed RoboFlow4D improves success rate for various robot manipulation tasks.

**Compliance With Llm Reviewing Policy:**

Affirmed.

**Final Justification:**

Thanks for the rebuttal. Since the author has already addressed my concerns. I have increased the score to Accept.

**Key Questions For Authors:**

1. VGGT is not in metric scale, will the unknown scale affect the 4D point track training? How do the authors address the scale problem in 3D space?
2. The authors claim the efficiency improvement is 120x, is this a fair comparison? (like the GPU number and type, task horizon, etc)
3. Most of the baselines are VLAs, how does the model perform compare with other 3D point trackers?
4. How accurate is the prediction of 4D point tracks? Some experiments compared with other point track baselines will be helpful.

**Limitations:**

Some discussions about failure modes will help to better understand the model design.

**Strengths And Weaknesses:**

Strength:

This paper proposes a highly efficient 4D point track predictor which enables robot planning. Real world experiments are important to understand the effectiveness and robustness of the model.

Weakness:

The experiments are mainly compared with VLA baselines, but lack comparison with other flow based 3D point track prediction methods.

---

> ### Author Rebuttal · Authors · 2026-03-31
>
> Thanks for your positive assessment of our work (e.g., ***high efficiency, practical planning capability, and real-world validation***), and constructive suggestions. Below, we respond to each point in detail.
>
> ---
> **W1. Comparison with 3D flow methods**
>
> Thanks for valuable comments. Below we compare RoboFlow4D with representative 3D flow methods, NovaFlow and Dream2Flow. RoboFlow4D enjoys those advantages: (1) **Goal-oriented objective**: Both NovaFlow and Dream2Flow predict fixed-horizon flows, while RoboFlow4D plans goal-oriented flows and yields higher adaptability to closed-loop manipulation. (2) **Lightweight end-to-end pipeline**: As a lightweight end-to-end framework, RoboFlow4D is more efficient than NovaFlow and Dream2Flow exploiting modular pipelines with stacked modules. (3) **Low-cost pure RGB input**: Both NovaFlow and Dream2Flow require RGB and depth inputs from extra depth estimation or more expensive RGBD cameras. In contrast, RoboFlow4D takes RGB-only inputs easily obtained from cheaper camaras.
>
> The advantages of RoboFlow4D are quantitatively shown in the table below (on L40): RoboFlow4D exhibits (1) lower flow prediction errors (**0.0142 vs 0.0354 and 0.0719**) and (2) sharply higher efficiency (e.g., **1/697 end-to-end latency**, **$<$ 1/45 GPU memory consumption**, and **< 1/2422 FLOPs), while (3) maintaining better manipulation performance ($>$ 2.8% higher SR**) compared to NovaFlow and Dream2Flow.
>
> |Method|3D Flow Error ↓|End-to-End Latency (s) ↓|Total GPU Memory (GB) ↓|Total FLOPs ↓|Success Rate (%) ↑|
> |-|-:|-:|-:|-:|-:|
> |NovaFlow|0.0719|596.20|35.33×4|35.22 PFLOPs|68.4|
> |Dream2Flow|0.0354|509.35|38.18×4|35.56 PFLOPs|69.2|
> |**RoboFlow4D (ours)**|**0.0142**|**0.73**|**3.12**|**14.54 TFLOPs**|**72.0**|
>
> ---
> **Q1. VGGT metric scale and effect**
>
> Thanks for constructive concerns. VGGT provides no direct metric scale but offers **3D geometric knowledge**. We inject this geometry awareness into RoboFlow4D by **only feature alignment**. Built upon 3D-aware RoboFlow4D, the action policy then **implicitly learns the metric scale in 3D space through 3D trajectory supervision**. Hence, the lack of VGGT's metric scale does not affect 4D point track training.
>
> ---
> **Q2. Efficiency improvement comparison**
>
> Thanks for the comment. 120x efficiency improvement is a **lowerbound** revealing **order-of-magnitude of system-level latency gap**. The table below fairly compares single-sample costs under minimum hardware requirements. Compared to 3D flow based NovaFlow and Dream2flow with modular pipelines requiring $\sim$ 10 minutes across 4 L40 GPUs, RoboFlow4D achieves **$>$ 697 inference speedup using only one single L40 GPU**. These will be appended to paper.
>
> |Method|GPU|Latency (s) ↓|
> |-|-|-|
> |NovaFlow|4×L40|596.20|
> |Dream2Flow|4×L40|509.35|
> |RoboFlow4D (ours)|**1×L40**|**0.73**|
>
> ---
> **Q3. Comparison with 3D flow methods**
>
> Thanks for pointing this out. The table in W1 compares the efficiency and success rate (SR) of RoboFlow4D with 3D flow competitors NovaFlow, Dream2Flow. RoboFlow4D yields **significantly higher efficiency and SR**. Details are in W1.
>
> ---
> **Q4. 4D point track accuracy**
>
> Thanks for the concern. **On one hand**, integrated with RoboFlow4D's flow plans, DP and DiT both obtain $>$ 5\% success rate gains (Tab. 1, main paper). **On the other hand**, RoboFlow4D yields a lower flow estimation error and a higher downstream manipulation success rate (72.0% vs 68.4% and 69.2%) over NovaFlow and Dream2Flow (table in W1). These consistent gains justify **RoboFlow4D's accurate flow plan to guide manipulaion**.
>
> **Limitation: failure modes**
>
> Thanks for valuable suggestions. We discuss failure modes and designs point by point.
> 1. **3D vs 2D**: 2D flow methods (e.g., Im2Flow2Act) produce plausible trajectories in 2D pixel space, but can not perform physically feasible in 3D space due to missing 3D geometry. Hence, RoboFlow4D plans **3D flows** as 3D priors for physical manipulation.
> 2. **End-to-end vs modular**: Existing 3D flow methods (e.g., Dream2Flow) adopt modular pipelines stacking multiple models (e.g., video generation) to predict 3D flows for manipulation. However, (1) they brings **huge computation costs** (table in W1), and (2) the video generation models often fail to predict reasonable videos, incurring **unsuccessful point tracking and manipulation** (https://anonymous.4open.science/w/RoboFlow4D/: Fig.B). Hence, we design a **lightweight end-to-end framework** to mitigate huge computation costs and flow prediction errors.
> 3. **Goal-oriented vs fixed-horizon**: Fixed-horizon flow methods (e.g., PointWorld) predict fixed-horizon flows across multiple subgoals or different manipulation phases, bringing execution difficulty and manipulation failures (https://anonymous.4open.science/w/RoboFlow4D/: Fig.C).
> Overall, RoboFlow4D benefits from designs: **3D spatial awareness, end-to-end flow prediction, goal-oriented adaptive planning**.

---

> > ### Author Rebuttal · Reviewer_nrX8 · 2026-04-03
> >
> > Thanks for the rebuttal response. I believe my concerns have been fully addressed. And the additional results are great to understand the comparison with other 3D-flow based baselines. I will raise the score to Accept.

---

> > > ### Author Response · Authors · 2026-04-03
> > >
> > > Thank you for your positive feedback and for recognizing that the additional results have addressed your concerns. We greatly appreciate your time and constructive comments, which have helped improve the clarity and completeness of the paper.

---

### Official Review · Reviewer_xKG4 · 2026-03-12

**Soundness:** 2
**Presentation:** 3
**Significance:** 3
**Originality:** 3
**Overall Recommendation:** 4
**Confidence:** 3

**Summary:**

This paper presents RoboFlow4D, a lightweight, end-to-end flow world model that directly predicts goal-oriented, multi-frame 3D trajectories from 2D RGB observations and language instructions. By serving as a low-frequency spatial planner, RoboFlow4D integrates seamlessly with general action policies to form a "slow-fast" closed-loop control system. Extensive evaluations across both simulated benchmarks and real-world manipulation tasks demonstrate that the framework improves task success rates while reducing computational latency compared to traditional modular pipelines.

**Compliance With Llm Reviewing Policy:**

Affirmed.

**Final Justification:**

The additional results in the rebuttal effectively support the paper's claims and clarify the advantages of the model's architecture. My concerns are addressed. I've raised my score.

**Key Questions For Authors:**

1. Could you clarify the origin of the optional input query points $\mathcal{Q}$? Are they manually sampled, or generated by another module? Additionally, is $n_{kp}$ always equal to $m$, and what specific physical locations do these keypoints track on the output flow?
2. What specific update frequency and $r$ were used for the main simulation and real-world experiments reported in Tables 1, 2, and 5?
3. Given that discarding components in Table 3 only increases the $l_2$ error by roughly 1-2 mm, can you provide the Success Rate for these ablated models? Demonstrating a drop in SR would convincingly prove the necessity of the Context Token, Query Points, and 3D Alignment modules.
4. Why are the overall success rates on ManiSkill3 so much lower than LIBERO? What specific types of failures is the base policy making in this single-view setting, and exactly what kind of physical prior is RoboFlow4D providing to DP and DiT to avoid these specific failures?
5. Regarding the recovery behavior shown in Figure 5, where does this capability originate? Are there similar recovery trajectories present in the training demonstrations, or is this behavior an emergent property of the closed-loop control? Do other VLA baselines exhibit similar recovery behaviors in your testing?

**Limitations:**

Yes.

**Strengths And Weaknesses:**

Pros:
1.  The paper is well-motivated to train a lightweight, end-to-end world model. By directly predicting goal-oriented flows from observations and instructions, the authors bypass the heavy computational overhead and accumulated errors typical of stacked modular pipelines.
2. The paper is clearly written, with adequate architectural details provided for the RoboFlow4D module, making the end-to-end pipeline easy to follow.
3. The experiments demonstrate clear performance gains when using RoboFlow4D as a prior for DP and DiT. The real-world experiments exhibit significantly lower latency compared to existing modular pipelines, demonstrating the practical applicability.

Cons:
1. The paper lacks a precise explanation of the IO data structures. It is unclear under what circumstances the optional input query points $\mathcal{Q} \in \mathbb{R}^{m \times 2}$  are necessary or where they originate during inference. Furthermore, regarding the output flow $\mathcal{F} \in \mathbb{R}^{n_{kp} \times K \times 3}$, the relationship between the number of input query points ($m$) and output keypoints ($n_{kp}$) is ambiguous.
2. While the slow-fast control system is a core contribution, the exact update frequencies used for the main results (Tables 1, 2, and 5) are not explicitly stated. The ablation study in Table 4 tests $r \in \{1, 2, 4\}$, which represent relatively minor frequency differences.
3. In Table 3, the ablation on the modular design relies on 3D $l_2$ error. The differences between the variants are on the scale of 1~2 mm. It is unconvincing that such a marginal difference in spatial error meaningfully impacts downstream manipulation tasks.

---

> ### Author Rebuttal · Authors · 2026-03-31
>
> Thank you for the careful reading and constructive feedback. Below, we address each point in detail in response to the insightful comments and suggestions.
>
> ---
>
> **W1/Q1. Query points & flow keypoints**
>
> Thanks for the constructive comment. The input query points $Q$ is randomly sampled from **gripper** grounding mask derived by Grounded-SAM2. We use $Q$ to extract point embedding as **gripper localization cues** to generate multi-frame gripper flows in 3D space. When $Q$ is optionally removed, the point embedding is set to zero, yielding higher flow generation error (**0.0158 m** vs **0.0142 m**, as shown in Tab. 3 in the main paper). Keypoints track of the output flow follow the same query points on gripper. Hence, $m=n_{\mathrm{kp}}$. We will append the explantions to the paper.
>
> ---
>
> **W2/Q2. Update ratio and frequency ablation**
>
> Thanks for pointing this out. The table below shows the success rate (SR) under various planner update ratios $r$. The SR reamins stable across different $r \in \{1,2,4\}$ (i.e., 71.0% - 72.0%), but sharply drops to 67.2% when $r$ further increases to 8. This drop stems from the mismatched flow planning for a very long executon horizon across multiple atomic tasks. **Overall, a larger update ratio $r$ yields higher efficiency from fewer planner updates, while incurring lower SR due to planning misalignment.** To make a better trade-off between efficiency and SR, **$r$ is set to 2** for Tab. 1, 2, and 5 of the paper. We will add the explanations in the camera-ready version.
>
> |Method|Setting|Success rate (%)|
> |-|-|-|
> |DP + RoboFlow4D|Fast/Slow *r*=8|67.2|
> |DP + RoboFlow4D|Fast/Slow *r*=4|71.0|
> |DP + RoboFlow4D|Fast/Slow *r*=2|71.8|
> |DP + RoboFlow4D|Fast/Slow *r*=1|**72.0**|
>
> ---
>
> **W3/Q3. L2 error & manipulation performance**
>
> Thanks for your valuable suggestions. The table below demonstrates the **consistent effectiveness** of each component in terms of **both 3D Error for flow prediction and success rate (SR) for the final manipulation**. As shown in the table, removing any component incurs higher 3D error and lower SR, which indicates that all components contribute to both the flow prediction and manipulation. The full table will be updated to the paper.
>
> | Method| 3D Error ↓ | Success Rate (%) ↑ |
> |-|-|-|
> | Full RoboFlow4D + DP |  **0.0142** |  **72.0**  |
> | w/o Context Token    |    0.0152   |     70.8   |
> | w/o Query Points     |    0.0158   |     70.4   |
> | w/o 3D Alignment     |    0.0160   |     70.2   |
>
> ---
>
> **Q4. Why ManiSkill3 is much lower than LIBERO and what RoboFlow4D provides? Specific types of failures the base policy is making in this single-view setting. Physical prior RoboFlow4D provides to DP and DiT to avoid these specific failures.**
>
> Thanks for raising the concern. The lower success rates on ManiSkill3 stems from its challenging benchmark settings: **heterogeneous simulation**, substantial **domain randomization** (as pointed out in [1] and [2]), and **single third-person view** that introduces weaker spatial cues, occlusion-caused failures, depth ambiguity, and more severe contact misalignment. **All baselines** suffer from these **common** challenges: **they are all substantially lower on ManiSkill3 than on LIBERO**.
>
> The base policy makes two mostly common failure modes: (1) **depth ambiguity** and (2) **dithering motions** once execution drifts away.
>
> To avoid these failures, RoboFlow4D offers an **explicit, goal-oriented multi-frame 3D flow plan** as strong geometric motion guidance for DP and DiT.
>
> References:
>
> [1] Tao S, Xiang F, Shukla A, et al. Maniskill3: Gpu parallelized robotics simulation and rendering for generalizable embodied ai. arXiv, 2024.
>
> [2] Yang X, Wu R, Liu J, et al. ReMAP-DP: Reprojected Multi-view Aligned PointMaps for Diffusion Policy. arXiv, 2026.
>
> ---
>
> **Q5. Where does the recovery behavior come from?**
>
> Thank you for the valuable concern. **Without any explicit supervision signal**, the recovery behavior is an **emergent ability** oringinated from: (1) **Closed-loop replanning**: At each planning step, RoboFlow4D predicts a new **goal-oriented** flow plan from the **current observation**, allowing the controller to adapt to updated motion guidance after execution drift and re-align with the task goal. (2) **Atomic-task supervision**: During training, the model learns to map to the same atomic goal from any arbitrary intermediate observation. The two synergistic designs empower RoboFlow4D with the capability of **goal-oriented corrective replanning**, rather than a single open-loop trajectory replay.
>
> As qualitatively compared in Figure A at https://anonymous.4open.science/w/RoboFlow4D/ side by side, RoboFlow4D exhibits **more reliable recovery after execution drift** over Pi0, which benefits from explicitly corrective guidance by a spatially grounded flow plan.

---

> > ### Author Rebuttal · Reviewer_xKG4 · 2026-04-02
> >
> > Thanks for the new experiments. The additional results support your claims and clarify the advantages of the model's architecture. While some of the ablation results were surprising, they successfully addressed my concerns. I will raise my score accordingly.

---

> > > ### Author Response · Authors · 2026-04-02
> > >
> > > Thank you for your feedback and for raising your score. We are glad that our responses addressed your concerns, and we deeply appreciate your kind recognition of our work.

---

### Decision · Program_Chairs · 2026-04-30

**Decision:**

Accept (regular)

**Comment:**

The submission introduced a lightweight flow world model that unifies perception and planning by estimating temporal motion in physical 3D space.  Reviewers liked the idea but raised concerns about its presentation and (mostly) experiments.  The authors did a good job during the rebuttal, after which all reviewers were convinced of the value of the work, and recommended Accept.  The AC agrees with the reviewers and would like to recommend acceptance.  The authors are encouraged to revise the submission based on the comments for the camera-ready.